# CLOSED-FORM DIFFUSION MODELS

## ABSTRACT

Score-based generative models (SGMs) sample from a target distribution by iteratively transforming noise using the score function of the perturbed target. For any finite training set, this score function can be evaluated in closed form, but the resulting SGM memorizes its training data and does not generate novel samples. In practice, one approximates the score by training a neural network via score-matching. The error in this approximation promotes generalization, but neural SGMs are costly to train and sample, and the effective regularization this error provides is not well-understood theoretically. In this work, we instead explicitly smooth the closed-form score to obtain an SGM that generates novel samples without training. We analyze our model and propose an efficient nearest-neighbor-based estimator of its score function. Using this estimator, our method achieves sampling times competitive with neural SGMs while running on consumer-grade CPUs.

## 1 INTRODUCTION

Score-based generative models (SGMs) draw samples from a target distribution $\rho_1$ by sampling Gaussian noise and flowing it through a possibly noisy velocity field $v_t$. This velocity field depends on the *score function* of the perturbed target distribution $\rho_t$, which existing SGMs parameterize as a neural network and learn via *score-matching* (Hyvärinen & Dayan, 2005) or denoising (Vincent, 2011; Ho et al., 2020). Although the target distribution $\rho_1$ (for example, the distribution over human face images) is in principle continuous, in practice score-matching and denoising problems are solved using an empirical approximation $\hat{\rho}_1$ to the target distribution constructed from a finite training set.

When $\hat{\rho}_1$ is the empirical distribution over a finite training set $\{x_i\}_{i=1}^N$, the perturbed target distribution $\rho_t$ is a mixture of Gaussians, whose score function $\nabla \log \rho_t(z)$ has a simple closed-form expression. This score function is a vector pointing from $z$ toward a distance-weighted average of all $N$ rescaled training points and is the *exact* solution to the score-matching problem (Miyasawa, 1961; Raphan & Simoncelli, 2011; Karras et al., 2022). By evaluating this *closed-form score* during sampling, one obtains a training-free sampler for $\hat{\rho}_1$. While this approach seems tempting at first glance, two flaws render it unsuitable for real-world applications: (1) Many applications involve large training sets, prohibiting $O(N)$ computation of the closed-form score, and (2) flowing base samples through the closed-form velocity field simply outputs training samples $x_i$, which is not useful in practice.

Existing work avoids these issues by neurally approximating the score of $\rho_t$. By compressing training data into the score model's weights, neural score functions replace a sum over $N$ training points with a neural network evaluation whose complexity does not depend directly on $N$. Moreover, neural SGMs generate novel samples given finite training data thanks to approximation error (from limited model capacity) and optimization error (from undertraining) in learning the score (Pidstrigach, 2022; Yoon et al., 2023; Yi et al., 2023). While neural SGMs are successful, they are costly to train, and sampling them requires many (typically GPU-bound) neural network evaluations. Furthermore, the form of the error that enables neural SGMs to generalize is unknown, making it difficult to characterize the distribution from which these models sample in practice.

Our key insight is that the flaws of naïve closed-form SGMs (in particular, lack of generalization and poor scalability) can be addressed without resorting to costly black-box neural approximations. To this end, we make use of a well-known score formula and introduce *smoothed closed-form diffusion*

*models* (smoothed CFDMs), a class of training-free diffusion models that require only access to the training set at sampling time. Smoothed CFDMs generate novel samples from a finite training set by flowing Gaussian noise through a velocity field built from a *smoothed* closed-form score. Our method is highly efficient, has few hyperparameters, and generates good-quality samples in high-dimensional tasks such as image generation. By developing this algorithm, we demonstrate that a closed-form score formula can be adapted to build a non-neural sampler that scales to non-trivial generative tasks.

*Our specific contributions are:* (1) In §4, we show that smoothing the exact solution to the score-matching problem promotes generalization. (2) Using our smoothed score, in §5.1 we construct a closed-form sampler that generates novel samples without requiring any training, and characterize the support of its samples. (3) In §5.2 and §5.3, we accelerate our sampler using a nearest-neighbor-based estimator of our smoothed score and by taking fewer sampling steps, with analysis and experiments showing that in practice, one can aggressively approximate our smoothed score at little cost to sample accuracy. (4) In §6, we scale our method to high-dimensional tasks such as image generation. By operating in the latent space of a pretrained autoencoder, we generate good-quality, novel samples from CIFAR-10 at a rate of 138 latents per second on a *consumer-grade laptop with no dedicated GPU*.

## 2 RELATED WORK

*Diffusion models* (Sohl-Dickstein et al., 2015; Song & Ermon, 2019; Ho et al., 2020) have recently achieved state of the art performance in image (Rombach et al., 2022; Zhang & Agrawala, 2023) and video generation (Ho et al., 2022a;b). They have also shown promise in 3D synthesis (Luo & Hu, 2021; Poole et al., 2022; Watson et al., 2022) and in crucial steps of the drug discovery pipeline such as molecular docking (Corso et al., 2023) and generation (Hoogeboom et al., 2022; Schneuing et al., 2022). Despite this progress, however, diffusion models remain costly to train and sample from (Shih et al., 2023). Prior work has sought to accelerate the sampling of diffusion models via model distillation (Salimans & Ho, 2022), operating in a pre-trained autoencoder's latent space (Vahdat et al., 2021; Rombach et al., 2022), modifying the generative process (Song et al., 2020), using alternative time discretizations for sampling (Zhang & Chen, 2023; Liu et al., 2022; Wu et al., 2023), or by parallelizing sampling steps (Shih et al., 2023). Latent diffusion models also benefit from lower training expenses (Rombach et al., 2022), but publicly-reported training costs for state-of-the-art diffusion models remain high (Bastian, 2022).

Recent works propose alternative diffusion-like models that discard the Markov chain and SDE formalisms from earlier work. Liu et al. (2023) introduce a unified framework for flow-based generative modeling that subsumes diffusion models and show that straightening their model's flows enables few-step sample generation. Heitz et al. (2023) use a similar objective to construct a straightforward graphics-inspired sampler, and Delbracio & Milanfar (2023) concurrently generalize this framework to arbitrary data perturbations and apply it to image restoration and generation tasks. All of these methods parametrize their flows by neural networks that require extensive training.

While diffusion models draw inspiration from mathematical theory (Feller, 1949; Stroock & Varadhan, 1969a;b; 1972), there have been limited attempts to develop a theoretical understanding. Salmona et al. (2022); Koehler et al. (2023) study the statistical limitations of diffusion models trained via score-matching, De Bortoli et al. (2021); Lee et al. (2023) present convergence results for diffusion models with absolutely continuous targets, and De Bortoli (2022) extends these results to manifold-supported distributions. However, as diffusion models are trained on an empirical approximation to their target distributions, these results can only show that a diffusion model converges to the empirical distribution of its training set, whereas one is typically interested in generating *novel* samples.

Pidstrigach (2022) takes an initial step in this direction by studying the support of an SGM's model distribution and providing conditions under which an SGM memorizes its training data or learns to sample from the true data manifold. Oko et al. (2023) further show that diffusion models can attain nearly minimax estimation rates for the true data distribution provided its density lies in an appropriate function class. Yoon et al. (2023) propose and empirically test a memorization-generalization dichotomy, which states that diffusion models may only generalize when they are parametrized by neural networks with insufficient capacity relative to the size of their training set. Yi et al. (2023) note that standard training objectives for diffusion models have closed-form optima given finite

training sets and show via experiments that the approximation error of neural score functions enables existing diffusion models to generalize. Whereas these works study the generalization of existing SGMs, we construct a novel SGM that explicitly perturbs the closed-form score to generalize without the indeterminate approximation error and training costs of a neural score.

Recent works in graphics and vision have also noted that neural networks are unnecessary for tasks such as novel view synthesis, where neural radiance fields (NeRFs) had previously achieved SOTA results (Barron et al., 2022). In light of this, Kerbl et al. (2023) use efficiently optimized 3D Gaussian scene representations to achieve SOTA visual quality in novel view synthesis while operating in real time. In this work, we adopt a similar perspective and investigate the extent to which neural networks can be replaced with efficient and well-understood classical approaches in generative modeling.

## 3 PRELIMINARIES: THE CLOSED-FORM SCORE

*Flow-based generative models* draw samples from a target distribution $\rho_1$ by sampling from a known base distribution $\rho_0$ (typically $\mathcal{N}(0, I)$) and flowing these samples through a velocity field $v_t$ from $t = 0$ to $t = 1$. For an appropriately-chosen $v_t$, the samples will be distributed according to the target distribution $\rho_1$ at $t = 1$. SGMs employ a $v_t$ that depends on the score function $\nabla \log \rho_t$ of the perturbed data distribution $\rho_t$. For example, when $\rho_0 = \mathcal{N}(0, I)$, one velocity field satisfying this property is $v_t^*(z) = \frac{1}{t}(z + (1-t)\nabla \log \rho_t^*(z))$ (Liu et al., 2023), where $\rho_t^*$ is the marginal distribution of the random variable $z = tx + (1-t)\epsilon$, whose samples are target samples $x \sim \rho_1$ that have been rescaled by $t$ and perturbed by Gaussian noise $(1-t)\epsilon \sim \mathcal{N}(0, (1-t)^2 I)$. The score function $\nabla \log \rho_t^*(z)$ is typically learned via score-matching or denoising.

In practice, one learns an SGM from a finite training set $\{x_i\}_{i=1}^N$. In this case, the target distribution $\hat{\rho}_1$ is the empirical distribution over $\{x_i\}_{i=1}^N$, and for the field $v_t^*$ defined above, the perturbed target distribution $\rho_t^*$ is a mixture of Gaussians with means $tx_i$ and common covariance matrix $(1-t)^2 I$. (We will use the fact that $\rho_t^*$ is a mixture of Gaussians to accelerate our sampler in §5.2.) Its score $\nabla \log \rho_t^*(z)$ has a closed-form expression:

$$\nabla \log \rho_t^*(z) = \frac{1}{(1-t)^2}(k_t(z) - z), \tag{1}$$

$$\text{where } k_t(z) = \sum_{i=1}^N \text{softmax}\left(-\frac{\|z - tX\|^2}{2(1-t)^2}\right)_i tx_i, \tag{2}$$

in which we let $\|z - tX\|^2$ denote the vector whose $i$-th entry is $\|z - tx_i\|^2$. This $\nabla \log \rho_t^*(z)$ is a vector pointing from $z$ toward a distance-weighted average $k_t(z)$ of all $N$ rescaled training points and is the *exact* solution to the score-matching problem given finite training data. Equation (1) is well-known, having appeared in the empirical Bayes literature as early as in Miyasawa (1961) and more recently in works such as Raphan & Simoncelli (2011) and Karras et al. (2022, Appendix B.3). It has also inspired machine learning methods such as denoising score-matching (Vincent, 2011) and score interpolation (Dieleman et al., 2022).

We define a *closed-form diffusion model* (CFDM) to be the SGM that flows $\mathcal{N}(0, I)$ base samples through this $v_t^*(z)$ while evaluating the score $\nabla \log \rho_t^*(z)$ in closed-form throughout the flow. As this model can only generate samples from the empirical distribution over training data, CFDMs are not useful in practice.

## 4 SMOOTHED CLOSED-FORM DIFFUSION MODELS

Pidstrigach (2022); Yi et al. (2023) find that existing diffusion models generalize due to approximation error incurred during score-matching. Rather than studying the generalization of neural SGMs, we take inspiration from this observation and *construct* a training-free SGM that generalizes by explicitly inducing error in the closed-form score.

### 4.1 DEFINITION

Deep neural networks fit the low-frequency components of their target functions first during training, a phenomenon known as "spectral bias" that results in excessively smooth approximations to the target function (Rahaman et al., 2019). Hence, to model the bias of a neural SGM, we induce error in the score function by *smoothing* it. To smooth a function $f$, one chooses a zero-mean noise

distribution $p_\epsilon$ and replaces $f(z)$ with the convolution $\tilde{f}(z) = \mathbb{E}_{\epsilon \sim p_\epsilon}[f(z + \epsilon)]$. In practice, we compute the smoothed score function $s_{\sigma,t}(z)$ by fixing a smoothing parameter $\sigma \geq 0$, drawing $M$ noise samples $\epsilon_m \sim p_\epsilon$, and computing

$$s_{\sigma,t}(z) = \frac{1}{(1-t)^2}\left(\frac{1}{M}\sum_{m=1}^{M} k_t(z + \sigma\epsilon_m) - z\right). \tag{3}$$

That is, we *average* the weights $k_t$ in (2) over $M$ small perturbations $\sigma\epsilon_m$ of $z$; as $\sigma \to 0$, we approach the unsmoothed score (1). We do not add noise to the $-z$ term in the score because it vanishes in expectation. The smoothing procedure in (3) is the key ingredient enabling our model to generalize without a learned approximation to the score function. The smoothed score $s_{\sigma,t}$ can then be inserted into an SGM sampling loop to yield a closed-form sampler that generates novel samples.



(a) Closed-form score    (b) Smoothed score

Figure 1: Effect of smoothing on the closed-form score (green arrows). Colors represent distance weights in $k_t(z)$; blue regions of space are drawn to the blue point on the left, and vice-versa.

To develop intuition for why this simple modification of the closed-form score (1) promotes generalization, we consider the behavior of the closed-form score as $t \to 1$. Figure 1 depicts the closed-form score (1) and its smoothed counterpart (3) at $t = 0.95$ for a simple case where the training data consists of two points $x_0$ (in blue) and $x_1$ (in red). In this regime, the temperature $(1-t)^2$ of the softmax in (2) is low, and $k_t(z)$ is effectively equal to the nearest neighbor of $z$ within the training set. Flowing points $z$ through a velocity field such as Liu et al. (2023)'s $v_t^*(z) = \frac{1}{t}(z + (1-t)\nabla\log\rho_t^*(z))$ causes them to flow towards their nearest training sample. As a result, an SGM based on this score function simply outputs training data.

On the other hand, the small perturbations $\sigma\epsilon_m$ in (3) occasionally push points $z$ near the Voronoi boundary between $x_0$ and $x_1$ into their neighboring Voronoi cell. Averaging $k_t$ over these perturbations yields a score function that instead points towards the line segment connecting $x_0$ and $x_1$. An SGM based on the *smoothed* score function will hence cause samples to flow towards weighted barycenters of the training points, which promotes generalization. We will make these intuitions rigorous in the following section by proving Proposition 4.1, which will enable us to constrain the support of our model's samples.

## 4.2 EFFECT OF SMOOTHING THE SCORE

In this section, we show that as $t \to 1$, the smoothed score points towards barycenters of these tuples rather than towards training points, thereby enabling our sampler to generalize. We first note that via a straightforward computation,

$$k_t(z + \sigma\epsilon_m) = \sum_{i=1}^{N}\text{softmax}\left(-\frac{\|z - tX\|^2 + \sigma t u_{i,m}}{2(1-t)^2}\right)_i tx_i, \tag{4}$$

where $u_{i,m} = -2\langle\epsilon_m, x_i\rangle$ is a scalar random variable. This shows that smoothing the score acts by perturbing the distance weights $-\|z - tx_i\|$, so one can directly add scalar noise $u_{i,m} \sim p_u$ to these weights instead of perturbing the inputs $z$ with noise $\epsilon_m \sim p_\epsilon$. To simplify our exposition, we will frame the remainder of our results from this perspective.

We now show that smoothing the closed-form score yields a function $s_{\sigma,t}(z)$ that points from $z$ towards a convex combination $k_{\sigma,t}(z)$ of *barycenters* $t\bar{c}_k = \frac{1}{M}\sum_{m=1}^{M}tx_{i(k,m)}$ of tuples $tC_k = (tx_{i(k,m)} : m = 1,...,M)$ of rescaled training points. The weights of this convex combination depend not only on the distance $\|z - t\bar{c}_k\|$ between $z$ and the barycenters $t\bar{c}_k$, but also on the *variance* of the tuples $tC_k$ and the noise terms $\bar{u}_k = \frac{1}{M}\sum_{m=1}^{M}u_{i(k,m)}$. Tuples consisting of tightly-clustered points have low variance and hence receive large weights in $k_{\sigma,t}(z)$, whereas tuples of distant points have high variance and receive small weights in $k_{\sigma,t}(z)$. We prove the following proposition in Appendix A.1.

**Proposition 4.1 ($s_{\sigma,t}$ points towards barycenters of training points)** *The smoothed score $s_{\sigma,t}(z)$ can be expressed as:*

$$s_{\sigma,t}(z) = \frac{1}{(1-t)^2} \left( k_{\sigma,t}(z) - z \right),$$ (5)

$$where \ k_{\sigma,t}(z) = \sum_{k=1}^{N^M} softmax \left( -\frac{M}{2(1-t)^2} \left( \|z - t\bar{c}_k\|^2 + Var(tC_k) + \sigma t \bar{u}_k \right) \right)_k t\bar{c}_k.$$ (6)

## 5 SAMPLING ALGORITHM

### 5.1 FORWARD EULER SCHEME FOR SAMPLING

Armed with the smoothed score $s_{\sigma,t}$, we are now in position to define our sampler. Following Liu et al. (2023), we draw $\mathcal{N}(0, I)$ base samples and flow them through

$$v_{\sigma,t}(z) = \frac{1}{t} \left( z + (1-t)s_{\sigma,t}(z) \right),$$ (7)

from $t = 0$ to $t = 1$. We discretize this ODE using a forward Euler scheme, leading to Algorithm 1 for sampling using the smoothed score.

---

**Algorithm 1** Sampling

**Require:** Training set $\{x_i\}_{i=1}^N$, noise $\{u_{i,m}\}$,
    step size $h = \frac{1}{S}$, initial sample $z_0 \sim \mathcal{N}(0, I)$
   **for** $n = 0, ..., S - 1$ **do**
      $t_n = \frac{n}{S}$
      $z_{n+1} \leftarrow z_n + h v_{\sigma,t_n}(z_n)$
   **end for**
   **return** $z_T$

---

The smoothed score (3) and Algorithm 1 jointly define our *smoothed closed-form diffusion model*; given a smoothing parameter $\sigma$, we call this a $\sigma$-CFDM. Using Algorithm 1, we can sample from a $\sigma$-CFDM given access only to the training data $x_i$ and noise samples. Notably, no training phase or neural network is needed for this procedure. By explicitly smoothing the closed-form score rather than relying on a neural network's approximation error, we can determine the support of our $\sigma$-CFDM's distribution. For sufficiently small step sizes, our model's samples will lie at the barycenters of tuples of training points.

**Theorem 5.1 (Support of $\sigma$-CFDM samples)** *All samples returned by Algorithm 1 are of form $z_S = \frac{S}{S-1} k_{\sigma,\frac{S-1}{S}}(z_{S-1})$. As the number of sampling steps $S \to \infty$ (equivalently, as the step size $\frac{1}{S} \to 0$), the model samples converge towards barycenters $z_S = \bar{c}_k$ of $M$-tuples of training points.*

We prove this theorem in Appendix A.2. While our sampler is easy to implement and training-free, it may be costly if the number of sampling steps $S$ and the number of training samples $N$ are large. We address these issues in the following sections. In §5.2 we show how to reduce sampling costs by taking fewer steps, and in §5.3 we show how to approximate our smoothed score using efficient nearest-neighbor search. This will permit our method to scale to high-dimensional real-world datasets while achieving sampling times competitive with existing methods and running on consumer-grade CPUs.

### 5.2 TAKING FEWER SAMPLING STEPS

As a CFDM's distribution $\rho_t^*$ is simply a time-dependent mixture of Gaussians centered at training points, it can be directly sampled at any time $t$ by uniformly sampling a mixture mean $tx_i$ and then sampling from a Gaussian centered at $tx_i$. We use this fact to sample a $\sigma$-CFDM with fewer steps by starting at $T > 0$ with samples from its corresponding unsmoothed CFDM. As a $\sigma$-CFDM does not have the same distribution as an unsmoothed CFDM, this approximation incurs some error, which we bound in the following theorem.

**Theorem 5.2 (Approximation error from starting at $T > 0$)** *Let $\rho_{1-\epsilon}^T$ be the model distribution at $t = 1 - \epsilon$ obtained by starting sampling a $\sigma$-CFDM at $T > 0$ with samples from the unsmoothed CFDM, and let $\rho_{\sigma,1-\epsilon}^0$ be the corresponding $\sigma$-CFDM model distribution when sampling starting at $T = 0$. Then,*

$$W_2(\rho_{\sigma,1-\epsilon}^0, \rho_{\sigma,1-\epsilon}^T) = O \left( \exp \left( \frac{1}{(1-T)^2} \right) \cdot \frac{\sigma}{\epsilon^2 (1-T)^2} \right).$$ (8)

*where $W_2$ is the 2-Wasserstein distance.*

Following De Bortoli (2022), we stop sampling at time $1 - \epsilon$ for some truncation parameter $\epsilon > 0$ to account for the fact that the smoothed score $s_{\sigma,t}$ blows up as $t \to 1$ due to division by $(1-t)^2$. We prove this theorem in Appendix A.3.

This result shows that initializing a $\sigma$-CFDM with samples from the unsmoothed CFDM $\rho_T^*$ at time $T > 0$ results in bounded error that increases with starting time $T$ and smoothing parameter $\sigma$, with linear dependence on $\sigma$. Intuitively, increasing $\sigma$ causes the unsmoothed velocity field $v_t^*$ to be a worse approximation to the smoothed velocity field $v_{\sigma,t}$ at any time $t$; Theorem 5.2 confirms that the cost to sample accuracy is linear in $\sigma$.

Note, however, that (8) is a loose bound, especially in its dependence on $T$. In §6.2, we show that one obtains highly accurate approximations to $\rho_{\sigma,1}$ while starting sampling at $T$ close to 1 and hence employing few sampling steps. In Appendix B, we further characterize the distribution of our model's samples drawn starting from the final step in Algorithm 1 when the weight perturbations $u_{i,m}$ in (4) are Gumbel-distributed.

### 5.3 Fast score computation via approximate nearest-neighbor search

In §5.2, we showed that by sampling a $\sigma$-CFDM starting at $T > 0$ with samples from an unsmoothed CFDM, we can reduce the cost of sampling a $\sigma$-CFDM by taking fewer steps. However, each step requires the evaluation of the smoothed score $s_{\sigma,t}(z)$ and hence a sum over $O(N)$ terms. For large datasets, each evaluation of $s_{\sigma,t}(z)$ is therefore costly and places substantial demands on memory.

In the $t \to 1$ regime, the temperature of the softmax in (4) is low, and the large sum is dominated by the handful of terms corresponding to the smallest values of $\|z - tx_i\|^2 - \sigma t u_{i,m}$. If $\sigma$ is sufficiently small, these terms correspond to the *nearest neighbors* of $z$ among the rescaled training points $tx_i$. This suggests that we can approximate the smoothed score $s_{\sigma,t}$ via an importance sampling-like scheme that subsamples terms in the $O(N)$ sum while ensuring that the nearest neighbors of $z$ are included with high probability.

Noting that the closed-form score $\nabla \log \rho_t^*(z) = \frac{\nabla \rho_t^*(z)}{\rho_t^*(z)}$ is the score of a Gaussian kernel density estimator (KDE) $\rho_t^*$, we employ Karppa et al. (2022)'s unbiased nearest-neighbor estimator for KDEs to estimate the denominator, and take its gradient to obtain an unbiased estimate of the numerator. We provide details on this estimator in Appendix C. Our estimator is computed using the $K$ nearest neighbors of $z$ in the training set and $L$ random samples from the remainder of the training set; we study the accuracy trade-offs associated with $K$ and $L$ in §6.2.

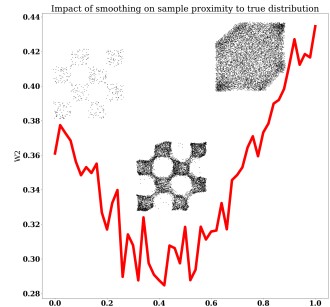

Given this estimator for the closed-form score, we estimate the smoothed score $\widehat{s_{\sigma,t}}$ via convolution against a smoothing kernel as in §4.1. By using the efficient approximate nearest neighbor search algorithms implemented in Faiss (Johnson et al., 2019), we are able to scale our method to high-dimensional real-world datasets and achieve sampling times competitive with neural SGMs while running on consumer-grade CPUs; see §6.3 and §6.4 for examples and runtime metrics.

Figure 2: $W_2$ between $\sigma$-CFDM model samples and true samples. We depict model samples for $\sigma \in \{0, 0.26, 1\}$.

## 6 Results

### 6.1 Impact of $\sigma$ on generalization

We now show that a $\sigma$-CFDM's model distribution approaches the true distribution $\rho_1$ of its training samples $x_i \sim \rho_1$ for appropriate values of $\sigma$. We fix a continuous target distribution $\rho_1$ and draw $N = 5000$ samples $y_i$ to serve as a discrete approximation to $\rho_1$. We then draw a smaller subset of $n = 500$ training samples $x_i$ and construct a $\sigma$-CFDM on these samples while varying $\sigma$.

For each $\sigma$, we measure the 2-Wasserstein distance $W_2$ between the $\sigma$-CFDM's generated samples and the true samples $y_i \sim \rho_1$, and use this as a tractable proxy for the distance between the $\sigma$-CFDM's model distribution and the true underlying distribution $\rho_1$ of the training samples. We present the results of this experiment for the "Checkerboard" distribution in Figure 2.

When $\sigma = 0$, the support of our model's samples (left side of Figure 2) coincides with the training samples $x_i$. The 2-Wasserstein distance between the model samples and true samples $y_i$ decreases for small values of $\sigma$ as the model samples become convex combinations of nearby points in the training set; we depict model samples for $\sigma = 0.26$ in the center of Figure 2. However, as $\sigma$ grows large, the model samples spread out to fill the convex hull of the training set (right side of Figure 2) and the distance between our model's samples and true samples $y_i$ grows rapidly. These results suggest that for appropriate values of $\sigma$, our method can use a fixed training set $\{x_i\}$ to generate novel samples that remain close to the target distribution $\rho_1$ from which the training samples were drawn.

In Figure 3, we demonstrate that with an appropriate choice of $\sigma$, our method can sample from a 2D surface embedded in $\mathbb{R}^3$ given a sparse blue noise sampling of the surface; this is a low-dimensional case of a manifold-supported distribution, which is typical in machine learning applications. Our method's samples (blue points) fill in the gaps between the sparse training samples (red points) while remaining close to the true manifold. This occurs because $\sigma$-CFDM samples are barycenters of tuples of nearby training points, with $\sigma$ controlling the variance of these tuples. For appropriate values of $\sigma$ and sufficiently dense samplings of training points, these barycenters will approximately lie on tangent planes to the surface, and hence lie near the surface but away from the training data.

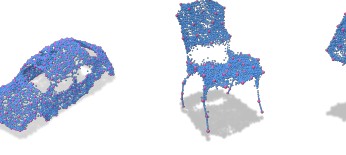

(a) $\sigma = 0.2$    (b) $\sigma = 0.375$    (c) $\sigma = 0.4$
28.9% ↓ in $W_2$    13.4% ↓ in $W_2$    34.1% ↓ in $W_2$

Figure 3: Sampling a $\sigma$-CFDM (blue points) yields a dense point cloud given sparse mesh samples (red points). We report % drop in $W_2$ distance to a dense mesh sampling when using our $\sigma$-CFDM's samples relative to the sparse training samples. We render these point clouds in Polyscope (Sharp et al., 2019).

## 6.2 ABLATION AND COMPUTATIONAL TRADE-OFFS

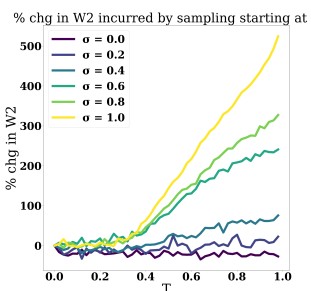

Figure 4: % change in $W_2$ between $\sigma$-CFDM model samples generated starting at $T = 0$ and samples generated starting at $T > 0$.

In this section, we investigate the impact of the start time $T$ and the parameters of our nearest-neighbor-based score estimator (39) on the distribution of our model's samples.

**Impact of $T$.** In §5.2, we showed that by sampling a smoothed CFDM starting at $T > 0$ with samples from an unsmoothed CFDM, we can reduce the cost of sampling a smoothed CFDM by taking fewer steps. However, the bound on approximation quality in Proposition 4.2.3 has exponential dependence on $(1 - T)$. In this section, we present empirical evidence that this bound is loose, and that for practical values of $\sigma$ one can begin sampling at $T$ close to 1 with little accuracy loss.

We fix a continuous target distribution $\rho_1$, draw $n = 500$ training samples $x_i$, and construct a $\sigma$-CFDM on these samples for smoothing parameters $\sigma \in \{0, 0.2, 0.4, 0.6, 0.8, 1.0\}$. We then vary the initial sampling times $T$ and compute the 2-Wasserstein distance $W_2$ between model samples generated starting at $T = 0$ and model samples generated starting at $T > 0$. We compare this to the average $W_2$ distance between batches of $\sigma$-CFDM samples generated by starting at $T = 0$ (which is nonzero due to randomness in sampling) and report the percent change in $W_2$ relative to this baseline value. We present the results of this experiment for the "Checkerboard" distribution in Figure 4.

For $\sigma < 0.4$, there is little accuracy loss when starting at $T > 0$, even for start times close to 1. When $\sigma \geq 0.4$, the accuracy of this approximation begins to decline for start times $T \geq 0.4$, with large reductions in approximation quality when both $\sigma$ and $T$ are large. As we have found that our model has yielded its best results for $\sigma \leq 0.4$ in the applications considered in this work, the results in this section support the use of few sampling steps in practice. The results in Sections 6.3 and 6.4 further support the use of late start times $T$ for image generation; we find in these experiments that we can start sampling as late as $T = 0.98$ while maintaining good sample quality.

**Impact of $K$ and $L$ on the NN-based score estimator.** In §5.3, we proposed an efficient score estimator based on fast nearest-neighbor search. We now study the impact of the number of nearest

neighbors $K$ and the number of random samples $L$ from the remainder of the training set on our model's samples.

We fix a continuous target distribution $\rho_1$, draw $n = 500$ training samples $x_i$, and construct a $\sigma$-CFDM on these samples for $\sigma = 0.3$; this value is typical for real-world applications. We then vary the number of nearest neighbors $K$ and the number of random samples $L$ used to compute the score estimator (39) and measure the 2-Wasserstein distance between model samples generated using the full smoothed score and model samples generated with the estimator (39). We present the results of this experiment for the "Checkerboard" distribution in Figure 5. We center the diverging color scheme at the $W_2$ distance between two batches of samples from a $\sigma$-CFDM using the full smoothed score, which represents a noise threshold encoding the inherent randomness in our model's samples across batches.

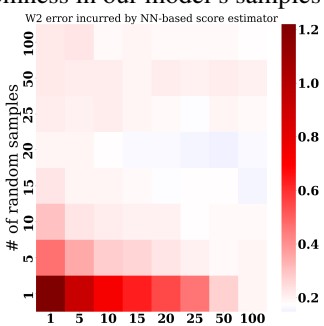

The error arising from the NN-based estimator is decreasing $K$ and $L$, with especially poor approximation quality arising from using a single random sample $x_\ell$. However, the accuracy of the model samples approaches the noise threshold for small values of $K, L$. For example, with $K = L = 15$ (which samples just 6% of the terms in $k_{\sigma,t}$), the $W_2$ distance between samples generated using the full score and the NN-based estimator is $0.1865$, a value close to the noise threshold of $0.1791$. In §6.3,6.4, we additionally show that one can generate high-quality images while subsampling $k_{\sigma,t}$ at a far lower rate, thereby enabling our method to scale to real-world datasets.

**Figure 5:** $W_2$ between $\sigma$-CFDM model samples generated using the full score and our NN-based estimator for varying # of NN $K$ (horizontal axis) and # of random samples $L$ (vertical axis).

### 6.3 IMAGE GENERATION IN PIXEL SPACE

In this section, we use our $\sigma$-CFDM to sample from the "Smithsonian Butterflies" dataset in pixel space[1]. Using our accelerated sampling techniques from Sections 5.2 and 5.3, we obtain in fast sampling times on consumer-grade CPUs. We benchmark our model's sample quality, training time, and sampling time against a Denoising Diffusion Probabilistic Model (DDPM) (Ho et al., 2020) and provide training details in Appendix D.1.

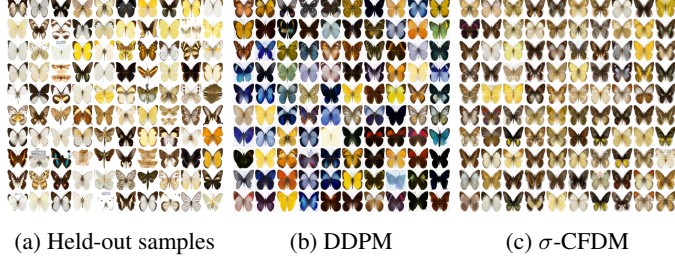

|  |  |  |
|:--:|:--:|:--:|
| (a) Held-out samples | (b) DDPM | (c) $\sigma$-CFDM |

We display images from a held-out test partition along with DDPM samples and our model's samples in Figure 6. Both our model and the DDPM (with 100 sampling steps) generate images that qualitatively resemble the test images, but as our model can only output barycenters of training samples (see Theorem 5.1), our samples exhibit somewhat softer details than the test and DDPM sam-

**Figure 6:** Held-out images from Smithsonian Butterflies (left), DDPM (center), and $\sigma$-CFDM samples (right) using start time $T = 0.98$, $M = 4$, step size $h = \frac{1}{S} = 10^{-2}$ and $\sigma = 0.3$

ples. Table 1 records sample quality metrics and training and generation times for our method and the DDPM baseline. Our method achieves comparable sample quality to a DDPM that has been trained for 2.38 hours without any training, and achieves a sample throughput of 5.83 samples/sec on a consumer-grade CPU – over 8 times the throughput of the DDPM running on a Tesla T4 GPU.

### 6.4 IMAGE GENERATION IN LATENT SPACE

Theorem 5.1 shows that in the limit of small step sizes, a $\sigma$-CFDM's samples are barycenters of nearby training points. This is a poor prior for images in pixel space, but an appropriately-chosen autoencoder may map the training data to a latent manifold that more closely satisfies this local linearity assumption. To this end, we follow Rombach et al. (2022) and sample from a $\sigma$-CFDM in the latent space of a pretrained autoencoder for the CIFAR-10 dataset. As in §6.3, we benchmark our model's sample quality, training time, and sampling time against a DDPM.

---

[1]Dataset available on Hugging Face: `huggan/smithsonian_butterflies_subset`

We display our model's samples, their nearest neighbors in the training set, and the first two PCA dimensions of the latent codes in Figures 7. In Figure 8, we display decoded samples from a DDPM and from a Gaussian fitted to the training latents. Operating in an autoencoder's latent space allows our method to generate plausible and diverse samples from CIFAR-10. The final column of Table 1 records metrics for our method and the DDPM baseline. Our method achieves comparable LPIPS to a DDPM that has been trained for 0.80 hours, and achieves a sample throughput of 138 latents/sec on a consumer-grade CPU, whereas the DDPM baseline's throughput is 13.5 latents/second on a Tesla T4 GPU. With a batch size of 25, the autoencoder decodes latents at a rate of 22.1k images/second on a T4 GPU.

| Method | Metric | Butterflies (128 x 128) | CIFAR-10 (latents) |
|--------|--------|-------------------------|--------------------|
| DDPM | LPIPS ↓ | 0.42 ± 0.00 | 0.39 ± 0.00 |
| | KID ↓ | 0.56 ± 0.01 | 0.02 ± 0.00 |
| | Training Time | 2.38 ± 0.02 h | 0.80 ± 0.05 h |
| | GPU (Tesla T4) Generation | 37.5 ± 0.43 s | 1.85 ± 0.27 s |
| $\sigma$-CFDM | LPIPS ↓ | 0.36 ± 0.00 | 0.41 ± 0.00 |
| | KID ↓ | 0.03 ± 0.00 | 0.02 ± 0.00 |
| | Training Time | 0 ± 0 h | 0 ± 0 h |
| | CPU (x86-64) Generation | 4.29 ± 0.94 s | 0.18 ± 0.05 s |

Table 1: Metrics for sample quality and generation time. GPU and CPU generation times are based on batches of 25 images. DDPM sampling time on CPU is approximately 2 hours for butterflies.

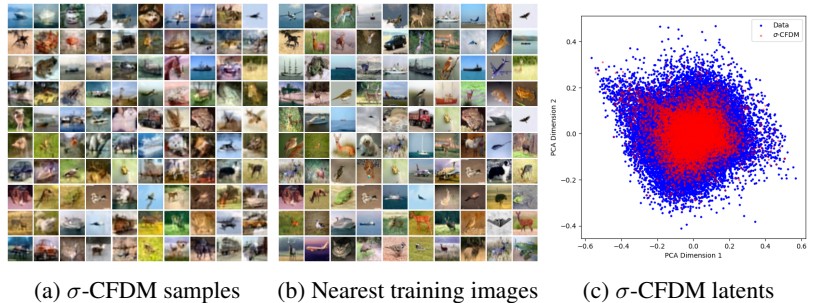

(a) $\sigma$-CFDM samples  (b) Nearest training images  (c) $\sigma$-CFDM latents

Figure 7: CIFAR-10 samples using a $\sigma$-CFDM with $T = 0.98$, $M = 2$, $h = 10^{-2}$, and $\sigma = 0.2$.

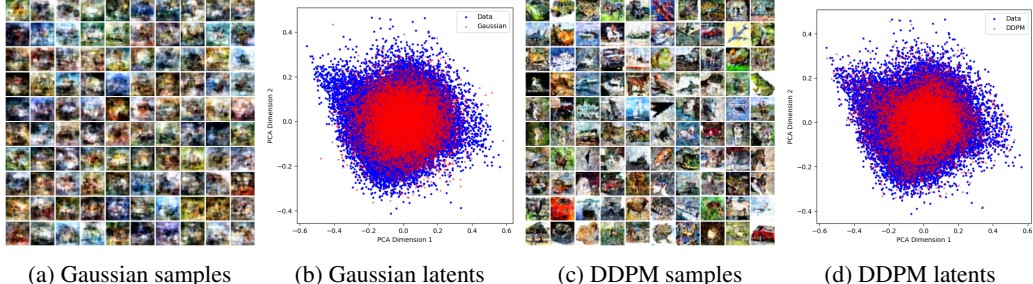

(a) Gaussian samples  (b) Gaussian latents  (c) DDPM samples  (d) DDPM latents

Figure 8: CIFAR-10 samples generated by sampling a multivariate Gaussian fitted to the training latents (a) and a DDPM in latent space (c).

## 7  CONCLUSION AND FUTURE WORK

In this work, we introduced smoothed closed-form diffusion models (smoothed CFDMs): a class of *training-free* diffusion models requiring only access to the training set at sampling time. Smoothed CFDMs leverage the availability of an exact solution to the score-matching problem—which alone does not yield generalization—and explicitly induce error by smoothing. Our results suggest that it is possible to design SGMs that generalize without relying on neural score approximations. However, to generate high-quality images, our method samples in the latent space of a pretrained autoencoder, which may be regularized to induce favorable latent structure. To this end, enforcing "local linearity," which requires training points to be representable as convex combinations of their nearest neighbors in latent space (Roweis & Saul, 2000; Du et al., 2021), may be an interesting avenue for exploration.

In future work, we will investigate whether large training sets and appropriate latent encodings can enable our method to achieve competitive sample quality in image generation. Using our model, we will also investigate whether smoothing is an adequate explanation for the generalization of neural SGMs, and whether employing alternative corruptions to the score function yields a closed-form SGM that generates high-quality images while operating directly in pixel space. Finally, we intend to apply our method to conditional generation tasks using Dhariwal & Nichol (2021)'s classifier guidance, which amounts to augmenting our velocity field (7) with the gradient of a pretrained classifier.

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

# A PROOFS

## A.1 PROOF OF PROPOSITION 4.1

For each $k = 1, ..., N^M$, let $tC_k = (tx_{i(k,m)} : m = 1, ..., M)$ be an $M$-tuple of rescaled training points $tx_i$. (The same point $tx_i$ can appear multiple times in an $M$-tuple.)

Define the barycenters and variances of these tuples as follows:

$$t\bar{c}_k = \frac{1}{M}\sum_{m=1}^{M} tx_{i(k,m)}, \quad \bar{u}_k = \frac{1}{M}\sum_{m=1}^{M} u_{i(k,m)}, \quad \mathrm{Var}(tC_k) = \frac{1}{M}\sum_{m=1}^{M} \|tx_{i(k,m)} - t\bar{c}_k\|^2. \quad (9)$$

We will show that up to a constant factor, the smoothed score (3) is itself the score of a mixture of $N^M$ Gaussians. Rewriting the smoothed score in gradient form, we have:

$$
\begin{aligned}
s_{\sigma,t}(z) &= \frac{1}{(1-t)^2}\frac{1}{M}\sum_{m=1}^{M}\sum_{i=1}^{N} \mathrm{softmax}\left(-\frac{\|z - tX\|^2 + \sigma t u_{i,m}}{2(1-t)^2}\right)_i (tx_i - z) \\
&= \nabla_z \frac{1}{M}\sum_{m=1}^{M}\log\sum_{i=1}^{N}\exp\left(-\frac{\|z - tx_i\|^2 + \sigma t u_{i,m}}{2(1-t)^2}\right) \\
&= \nabla_z \frac{1}{M}\log\prod_{m=1}^{M}\sum_{i=1}^{N}\exp\left(-\frac{\|z - tx_i\|^2 + \sigma t u_{i,m}}{2(1-t)^2}\right) \\
&= \nabla_z \frac{1}{M}\log\sum_{k=1}^{N^M}\exp\left(-\frac{\sum_{m=1}^{M}(\|z - tx_{i(k,m)}\|^2 + \sigma t u_{i(k,m)})}{2(1-t)^2}\right) \\
&= \nabla_z \frac{1}{M}\log\sum_{k=1}^{N^M}\exp\left(-\frac{M\left(\|z - t\bar{c}_k\|^2 + \mathrm{Var}(tC_k) + \sigma t \bar{u}_k\right)}{2(1-t)^2}\right) \qquad (*) \\
&= \frac{1}{M}\nabla_z \log\sum_{k=1}^{N^M}\exp\left(-\frac{M(\mathrm{Var}(tC_k) + \sigma t \bar{u}_k)}{2(1-t)^2}\right)\exp\left(-\frac{M\|z - t\bar{c}_k\|^2}{2(1-t)^2}\right) \\
&= \frac{1}{M}\nabla_z \log\sum_{k=1}^{N^M} w_k(t)\exp\left(-\frac{M\|z - t\bar{c}_k\|^2}{2(1-t)^2}\right) \\
&= \frac{1}{M}\nabla_z \log q_t(z)
\end{aligned}
$$

This shows that up to a constant factor $\frac{1}{M}$, the smoothed score $s_{\sigma,t}(z)$ is the score of a large mixture of Gaussians $q_t(z) = \sum_{k=1}^{N^M} w_k(t)\exp\left(-\frac{M(\|z - t\bar{c}_k\|^2)}{2(1-t)^2}\right)$. The mean of each Gaussian

is the barycenter $t\bar{c}_k$ of some $M$-tuple $t\tilde{C}_k$ of training points $tx_{i(k,m)}$, and its common covariance matrix is $\frac{(1-t)^2}{M}I$. The time-dependent mixture weights $w_k(t) \propto \exp\left(-\frac{M(\text{Var}(tC_k)+\sigma t\bar{u}_k)}{2(1-t)^2}\right)$ are decreasing in the variance of the $M$-tuples $tC_k$ but are subject to the presence of noise terms $\sigma t\bar{u}_k = \frac{\sigma t}{M}\sum_m u_{i(k,m)}$.

Finally, by expanding the gradient in $(*)$, we straightforwardly obtain:

$$s_{\sigma,t}(z) = \frac{1}{(1-t)^2}\left(\sum_{k=1}^{N^M}\text{softmax}\left(-\frac{M}{2(1-t)^2}\left(\|z-t\bar{c}_k\|^2 + \text{Var}(tC_k) + \sigma t\bar{u}_k\right)\right)_k t\bar{c}_k - z\right)$$

## A.2 PROOF OF THEOREM 5.1

Define

$$\tilde{k}_{\sigma,t}(z) = \sum_{k=1}^{N^M}\text{softmax}\left(-\frac{M}{2(1-t)^2}\left(\|z-t\bar{c}_k\|^2 + \text{Var}(t\tilde{C}_k) + \sigma t\bar{u}_k\right)\right)_k t\bar{c}_k \qquad (10)$$

so that $s_{\sigma,t}(z) = \frac{1}{(1-t)^2}(\tilde{k}_{\sigma,t}(z) - z)$. Then

$$\begin{aligned}
v_{\sigma,t}(z) &= \frac{1}{t}\left(z + (1-t)s_{\sigma,t}(z)\right) \\
&= \frac{1}{t}\left(z + \frac{1}{(1-t)}(\tilde{k}_{\sigma,t}(z) - z)\right) \\
&= \frac{1}{1-t}\left(\frac{1}{t}\tilde{k}_{\sigma,t}(z) - z\right)
\end{aligned}$$

Expanding the formula for the final Euler step using this expression for $v_{\sigma,t}(z)$ and $t_{S-1} = \frac{S-1}{S}$, we obtain:

$$\begin{aligned}
z_S &= z_{S-1} + \frac{1}{S}v_{\sigma,t_{S-1}}(z_{S-1}) \\
&= z_{S-1} + \frac{1}{S}\cdot\frac{1}{1-\frac{S-1}{S}}\left(\frac{1}{\frac{S-1}{S}}\tilde{k}_{\sigma,\frac{S-1}{S}}(z_{S-1}) - z_{S-1}\right) \\
&= z_{S-1} + \frac{S}{S-1}\tilde{k}_{\sigma,\frac{S-1}{S}}(z_{S-1}) - z_{S-1} \\
&= \frac{S}{S-1}\tilde{k}_{\sigma,\frac{S-1}{S}}(z_{S-1}) \\
&= \frac{S}{S-1}\sum_{k=1}^{N^M}\text{softmax}\left(-\frac{MS^2}{2}\left(\|z_{S-1}-\frac{S-1}{S}\bar{c}_k\|^2 + \text{Var}(\frac{S-1}{S}\tilde{C}_k) + \sigma\frac{S-1}{S}\bar{u}_k\right)\right)_k\frac{S-1}{S}\bar{c}_k \\
&\xrightarrow[S\to\infty]{} \bar{c}_{k^*}
\end{aligned}$$

In the final line, we use the fact that as $S \to \infty$, the temperature of the softmax goes to 0 and picks out a single index $k^*$ such that

$$\begin{aligned}
k^* &= \underset{k}{\text{argmax}} - \left(\|z_{S-1}-\bar{c}_k\|^2 + \text{Var}(\tilde{C}_k) + \sigma\bar{u}_k\right) \\
&= \underset{k}{\text{argmin}}\left(\|z_{S-1}-\bar{c}_k\|^2 + \text{Var}(\tilde{C}_k) + \sigma\bar{u}_k\right)
\end{aligned}$$

A.3    PROOF OF THEOREM 5.2

We divide the proof of this theorem into three propositions. We first sketch the proof and state the propositions, and then prove each proposition in subsections below.

Our first result shows that flowing $\rho_0$ through two similar velocity fields $v_t^*, v_{\sigma,t}$ yields similar model distributions $\rho_T^*, \rho_{\sigma,T}$ at some terminal time $T$:

**Proposition A.1** *Suppose a measure $\rho_0$ is pushed through velocity fields $v_t^*, v_{\sigma,t}$, and denote the respective pushforward measures at time $t$ by $\rho_t^*, \rho_{\sigma,t}$. Then,*

$$W_2(\rho_T^*, \rho_{\sigma,T}) \leq \int_0^T \beta(t) \sqrt{\underset{z \sim \rho_t^*}{\mathbb{E}} \|v_t^*(z) - v_{\sigma,t}(z)\|^2} dt \tag{11}$$

*where $\beta(t) := \exp\left(\int_t^T L_{v_s^*} ds\right)$ and $L_{v_s^*} \geq 0$ is the Lipschitz constant of $v_s^*$.*

The result above applies to any two velocity fields, subject to some weak regularity conditions. To apply this result to the unsmoothed and smoothed velocity fields $v_t^*$ and $v_{\sigma,t}$, we bound $\beta(t)$ and $\mathbb{E}\|v_t^*(z) - v_{\sigma,t}(z)\|^2$ in terms of $\sigma$:

**Proposition A.2** *Let $v_t^*$ be velocity field of an unsmoothed CFDM, and let $v_{\sigma,t}^*$ be the velocity field (7) of the corresponding $\sigma$-CFDM. Then,*

$$\beta(t) \leq \exp\left(\frac{C_0}{(1-T)^2}\right) \cdot \frac{1-t}{1-T} \tag{12}$$

*and*

$$\sqrt{\underset{z \sim \rho_t^*}{\mathbb{E}} \|v_t^*(z) - v_t(z)\|^2} \leq C_1 \frac{\sigma t}{2(1-t)^3} \tag{13}$$

*where $C_0, C_1$ are constants depending on the training data and the distribution $p_u$ of the scalar noise $u_{i,m}$ perturbing the distance weights in (4).*

Combining these results, we obtain the following bound on $W_2(\rho_T^*, \rho_{\sigma,T})$:

$$W_2(\rho_T^*, \rho_{\sigma,T}) = O\left(\exp\left(\frac{1}{(1-T)^2}\right) \cdot \frac{\sigma}{(1-T)^2}\right). \tag{14}$$

This shows that one can approximate a $\sigma$-CFDM's model samples at some time $T > 0$ by model samples from its corresponding unsmoothed CFDM (i.e. a mixture of Gaussians) when $T$ is sufficiently close to 0 and the smoothing parameter $\sigma$ is small.

We now show that flowing two similar distributions $\rho_T^*$ and $\rho_{\sigma,T}$ through a $\sigma$-CFDM's velocity field from time $T$ to $1 - \epsilon$ yields similar terminal distributions $\rho_{\sigma,1-\epsilon}^T, \rho_{\sigma,1-\epsilon}^0$. Following De Bortoli (2022), we stop sampling at time $1 - \epsilon$ for some truncation parameter $\epsilon > 0$ to account for the fact that the smoothed score $s_{\sigma,t}$ blows up as $t \to 1$ due to division by $(1-t)^2$.

**Proposition A.3** *Suppose $\rho_T^*$ and $\rho_{\sigma,T}$ are pushed through the velocity field $v_{\sigma,t}$ of a $\sigma$-CFDM, and let $\rho_{\sigma,1-\epsilon}^T, \rho_{\sigma,1-\epsilon}^0$ denote their respective terminal distributions at time $1 - \epsilon$. Then*

$$W_2(\rho_{\sigma,1-\epsilon}^T, \rho_{\sigma,1-\epsilon}^0) \leq O\left(\frac{1}{\epsilon^2}\right) W_2(\rho_T^*, \rho_{\sigma,T}). \tag{15}$$

By combining (14) and (15), we finally obtain a global upper bound on $W_2(\rho_{\sigma,1-\epsilon}^T, \rho_{\sigma,1-\epsilon}^0)$:

$$W_2(\rho_{\sigma,1-\epsilon}^T, \rho_{\sigma,1-\epsilon}^0) = O\left(\exp\left(\frac{1}{(1-T)^2}\right) \cdot \frac{\sigma}{\epsilon^2(1-T)^2}\right) \tag{16}$$

where $\rho_{\sigma,1-\epsilon}^T$ is the model distribution obtained by starting sampling at $T > 0$ with samples from the unsmoothed CFDM and $\rho_{\sigma,1-\epsilon}^0$ is true model distribution of the $\sigma$-CFDM.

### A.3.1 PROOF OF PROPOSITION A.1

Our proof for this proposition employs techniques similar to those used to prove Theorem 1 and Proposition 1 in Kwon et al. (2022).

We begin with the following well-known result (Santambrogio, 2015, Corollary 5.25):

Suppose that two measures $\rho^*$ and $\rho$ are each pushed through velocity fields $v_t^*, v_t$ respectively and denote the pushforward measures at time $t$ by $\rho_t^*, = \rho_t^*$. Then:

$$\frac{1}{2}\frac{d}{dt}W_2^2(\rho_t^*, \rho_t) = \mathbb{E}_{(x,y)\sim\gamma_t}\langle y - x, v_t^*(y) - v_t(x)\rangle \tag{17}$$

where $\gamma_t$ is the $W_2$ coupling between $\rho_t^*$ and $\rho_t$.

For any $x, y$ we can use Cauchy-Schwarz and the triangle inequality to obtain the following bound:

$$\langle y - x, v_t^*(y) - v_t(x)\rangle \leq \|y - x\| \cdot (\|v_t^*(y) - v_t^*(x)\| + \|v_t^*(x) - v_t(x)\|) \tag{18}$$

We can then bound $\|v_t^*(y) - v_t^*(x)\|$ in terms of maximum of the Jacobian $Dv_t^*$ of $v_t^*$ on the line segment $[x, y] := \{ty + (1-t)x : 0 \leq t \leq 1\}$ to obtain:

$$\langle y - x, v_t^*(y) - v_t(x)\rangle \leq \left(\max_{p\in[x,y]}\|Dv_t^*\|\right)\|y - x\|^2 + \|y - x\| \cdot \|v_t^*(x) - v_t(x)\| \tag{19}$$

This constant is in turn upper-bounded by the Lipschitz constant $L_{v_t^*}$ of $v_t^*$ on the convex hull of $\text{supp}(\rho_t^*) \cup \text{supp}(\rho_t)$, so we in fact have:

$$\langle y - x, v_t^*(y) - v_t(x)\rangle \leq L_{v_t^*}\|y - x\|^2 + \|y - x\| \cdot \|v_t^*(x) - v_t(x)\| \tag{20}$$

Adding $\mathbb{E}_{(x,y)\sim\gamma_t}$ back in, we get:

$$\begin{aligned}
\frac{1}{2}\frac{d}{dt}W_2^2(\rho_t^*, \rho_t) &= \mathbb{E}_{(x,y)\sim\gamma_t}\langle y - x, v_t^*(y) - v_t(x)\rangle \\
&\leq L_{v_t^*}\mathbb{E}_{(x,y)\sim\gamma_t}\|y - x\|^2 + \mathbb{E}_{(x,y)\sim\gamma_t}\|y - x\| \cdot \|v_t^*(x) - v_t(x)\| \\
&= L_{v_t^*}W_2^2(\rho_t^*, \rho_t) + \mathbb{E}_{(x,y)\sim\gamma_t}\|y - x\| \cdot \|v_t^*(x) - v_t(x)\| \\
&\leq L_{v_t^*}W_2^2(\rho_t^*, \rho_t) + \sqrt{\mathbb{E}_{(x,y)\sim\gamma_t}\|y - x\|^2} \cdot \sqrt{\mathbb{E}_{(x,y)\sim\gamma_t}\|v_t^*(x) - v_t(x)\|^2} \\
&= L_{v_t^*}W_2^2(\rho_t^*, \rho_t) + W_2(\rho_t^*, \rho_t) \cdot \sqrt{\mathbb{E}_{x\sim\rho_t^*}\|v_t^*(x) - v_t(x)\|^2}
\end{aligned}$$

where we use Cauchy-Schwarz for random variables in passing from the third to fourth lines and then the fact that $\rho_t^*$ is one of the marginals of $\gamma_t$. Using the chain rule on the LHS and cancelling a factor of $W_2(\rho_t^*, \rho_t)$ from both sides, we obtain the following differential inequality:

$$\frac{d}{dt}W_2(\rho_t^*, \rho_t) \leq L_{v_t^*}W_2(\rho_t^*, \rho_t) + \sqrt{\mathbb{E}_{x\sim\rho_t^*}\|v_t^*(x) - v_t(x)\|^2} \tag{21}$$

We can now solve the differential inequality (21) to obtain:

$$W_2(\rho_T^*, \rho_T) \leq \int_0^T \beta(t) \sqrt{\mathop{\mathbb{E}}_{x \sim \rho_t^*} \|v_t^*(x) - v_t(x)\|^2} \tag{22}$$

where

$$\beta(t) := \exp\left(\int_t^T L_{v_s^*} \mathrm{d}s\right) \tag{23}$$

### A.3.2    PROOF OF PROPOSITION A.2

We first estimate $\beta(t) = \exp\left(\int_t^T L_{v_s^*} \mathrm{d}s\right)$. As

$$v_t^*(z) = \frac{1}{t(1-t)} k_t^*(z) - \frac{1}{1-t} z, \tag{24}$$

we can bound its Lipschitz constant by $L_{v_t^*} \leq \max\left\{\frac{1}{t(1-t)} L_{k_t}, \frac{1}{1-t}\right\}$. Our next step is therefore to bound $L_{k_t^*}$.

If $X \in \mathbb{R}^{D \times N}$ is the matrix of training data and $w(z) = \mathrm{softmax}\left(-\frac{\|z - tx\|^2}{2(1-t)^2}\right) \in \mathbb{R}^N$ is the vector of weights, then $k_t^*(z) = tXw^*(z)$, so $L_{k_t} \leq t\|X\| L_{w^*(z)}$.

This softmax function is $\frac{1}{2(1-t)^2}$-Lipschitz with respect to $\|z - tx_i\|^2$. If $\|z - tx_i\| \leq A$ for all $z$ and all $tx_i$, then $\|z - tx_i\|^2$ is $2A$-Lipschitz with respect to $z$ and we can conclude that:

$$\begin{aligned}
L_{k_t^*} &\leq t\|X\| L_{w^*(z)} \\
&\leq t\|X\| \frac{2A}{2(1-t)^2} \\
&= t\|X\| \frac{A}{(1-t)^2}
\end{aligned}$$

Hence $L_{v_t^*} \leq \max\left\{\frac{A\|X\|}{(1-t)^3}, \frac{1}{1-t}\right\}$.

We now use this bound on $L_{v_t^*}$ to estimate $\beta(t)$. Let $\bar{s} \in [0, T]$ denote the time from which $\frac{A\|X\|}{(1-\bar{s})^3} \geq \frac{1}{1-\bar{s}}$. Then $\bar{s} = \max\{0, 1 - \sqrt{A\|X\|}\}$. Decomposing the integral that defines $\log \beta(t)$, we obtain:

$$\begin{aligned}
\int_t^T L_{v_s^*} \mathrm{d}s &= \int_t^{\bar{s}_t} L_{v_s^*} \mathrm{d}s + \int_{\bar{s}}^T L_{v_s^*} \mathrm{d}s \\
&\leq \int_t^{\bar{s}} \frac{1}{1-s} \mathrm{d}s + A\|X\| \int_{\bar{s}}^T \frac{1}{(1-s)^3} \mathrm{d}s \\
&= \log\left(\frac{1-t}{1-\bar{s}}\right) + \frac{A\|X\|}{2}\left(\frac{1}{(1-T)^2} - \frac{1}{(1-\bar{s})^2}\right) \\
&\leq \log\left(\frac{1-t}{1-T}\right) + \left(\frac{A\|X\|}{2(1-T)^2} - \frac{1}{2}\right)
\end{aligned}$$

Substituting this bound into $\beta(t) = \exp(\int_t^T L_{v_s^*})$ and simplifying, we obtain:

$$\beta(t) \leq C(T) \cdot \frac{1-t}{1-T} \tag{25}$$

where $C(T) = \exp\left(\frac{A\|X\|}{2(1-T)^2} - \frac{1}{2}\right)$.

We now estimate $\sqrt{\mathbb{E}_{x\sim\rho_t^*}\|v_t^*(x) - v_t(x)\|^2}$.

We first observe that $v_t^*(z) - v_t(z) = \frac{1}{t(1-t)}(k_t^*(z) - k_t(z))$. Once again letting $X \in \mathbb{R}^{D\times N}$ be the matrix of training data and $w^*(z) = \text{softmax}\left(-\frac{\|z-tx\|^2}{2(1-t)^2}\right) \in \mathbb{R}^N$, $\tilde{w}_m(z) = \text{softmax}\left(-\frac{\|z-tx\|^2 + \sigma t u_{i,m}}{2(1-t)^2}\right) \in \mathbb{R}^N$ be the vector of weights, we have that $k_t^*(z) = tXw^*(z)$ and hence

$$\|v_t^*(z) - v_t(z)\| = \frac{1}{1-t}\|X(w^*(z) - \frac{1}{M}\sum_{m=1}^M \tilde{w}_m(z))\| \le \frac{1}{1-t}\|X\| \cdot \frac{1}{M}\sum_{m=1}^M \|w^*(z) - \tilde{w}_m(z)\|. \tag{26}$$

Once again using the Lipschitz continuity of $w(z)$, we obtain the bound

$$\|w^*(z) - \tilde{w}_m(z)\| \le \frac{\sigma t u_{i,m}}{2(1-t)^2}, \tag{27}$$

and by substituting this into our bound for $\|v_t^*(z) - v_t(z)\|$, we obtain the bound:

$$\|v_t^*(z) - v_t(z)\|^2 \le \frac{t^2\sigma^2\|X\|^2\bar{u}_i^2}{4(1-t)^6}, \tag{28}$$

where $\bar{u}_i = \frac{1}{M}\sum_m u_{i,m}$. As this bound holds for all $z$, it also holds in expectation, so we finally conclude that

$$\sqrt{\mathbb{E}_{x\sim\rho_t^*}\|v_t^*(x) - v_t(x)\|^2} \le \frac{t\sigma\|X\|\bar{u}_i}{2(1-t)^3}. \tag{29}$$

### A.3.3  PROOF OF PROPOSITION A.3

We now begin with the differential inequality

$$\frac{\mathrm{d}}{\mathrm{d}t}W_2(\rho_t^*, \rho_t) \le L_{v_t^*}W_2(\rho_t^*, \rho_t) + \sqrt{\mathbb{E}_{x\sim\rho_t^*}\|v_t^*(x) - v_t(x)\|^2}, \tag{30}$$

that we derived in the proof of Proposition A.1, which bounds the rate of change in $W_2(\rho_t^*, \rho_t)$ when flowing $\rho_t^*$ and $\rho_t$ through two velocity fields $v_t^*$ and $v_t$, respectively. As we now consider the case where $\rho_T^*$ and $\rho_{\sigma,t}$ both flow through the smoothed velocity field $v_{\sigma,t}$ from time $T$ to $1-\epsilon$, $\sqrt{\mathbb{E}_{x\sim\rho_t^*}\|v_t^*(x) - v_t(x)\|^2} = 0$ and the differential inequality becomes:

$$\frac{\mathrm{d}}{\mathrm{d}t}W_2(\rho_t^*, \rho_{\sigma,t}) \le L_{v_{\sigma,t}}W_2(\rho_t^*, \rho_{\sigma,t}). \tag{31}$$

Solving this differential inequality, we obtain

$$W_2(\rho_{\sigma,1-\epsilon}^T, \rho_{\sigma,1-\epsilon}^0) := W_2(\rho_{1-\epsilon}^*, \rho_{\sigma,1-\epsilon}) \le \tilde{\beta}(T)W_2(\rho_T^*, \rho_{\sigma,T}) \tag{32}$$

where $\tilde{\beta}(T) = \int_T^{1-\epsilon} L_{v_{s,\sigma}}\mathrm{d}s$. Using the same bounds as in our proof of Proposition A.2 while noting that $v_{\sigma,t}$ is at least as smooth as $v_t^*$, we obtain

$$\tilde{\beta}(T) \le \log(\frac{1}{\epsilon}) + \frac{A\|X\|}{2\epsilon^2} - \frac{1}{2} = O\left(\frac{1}{\epsilon^2}\right). \tag{33}$$

Substituting this into (32), we obtain

$$W_2(\rho_{\sigma,1-\epsilon}^T, \rho_{\sigma,1-\epsilon}^0) \leq O\left(\frac{1}{\epsilon^2}\right) W_2(\rho_T^*, \rho_{\sigma,T}). \tag{34}$$

# B DISTRIBUTION OF ONE-STEP SAMPLES UNDER GUMBEL WEIGHT PERTURBATIONS

When the scalar noise $u_{i,m}$ perturbing the distance weights in (4) is drawn from a Gumbel$(0,1)$ distribution, we can precisely characterize the smoothed model's distribution when performing *single-step sampling* by starting sampling at the final Euler iteration in Algorithm 1.

**Proposition B.1** *Suppose we begin sampling a smoothed CFDM at iteration $S-1$ of Algorithm 1 using samples $z_{S-1} \sim \rho_{t_{S-1}}^*$ from the unsmoothed CFDM at $t_{S-1}$. Suppose also that the perturbations $u_{i,m}$ to the distance weights in (4) are drawn from a Gumbel$(0,1)$ distribution. Then, as the number of Euler steps $S \to \infty$, the model samples $z_S$ are of the form $z_S = \frac{1}{M} X I_\sigma$, where $X \in \mathbb{R}^{D \times N}$ is the matrix whose $i$-th column is training sample $x_i$ and $I_\sigma \sim Multinomial(\pi_\sigma, M)$. The probability $\pi_\sigma^i$ of training point $x_i$ is given by $\pi_\sigma^i = softmax\left(-\frac{1}{\sigma}\|z_{S-1} - x_i\|^2\right)$.*

The proof is as follows. We showed in Theorem 5.1 (see Appendix A.2) that as the number of sampling steps $S \to \infty$, the samples from a smoothed CFDM converge towards barycenters $z_S = \bar{c}_{k^*}$ of $M$-tuples of training points for indices $k^*$ such that:

$$k^*(z_{S-1}) = \underset{k}{\operatorname{argmax}} - \left(\|z_{S-1} - \bar{c}_k\|^2 + \operatorname{Var}(\tilde{C}_k) + \sigma \bar{u}_k\right) \tag{35}$$

Using an equivalent expression for $\tilde{k}_{\sigma,t}$, these barycenters can also be written as

$$z_S = \bar{c}_{k^*} = \frac{1}{M} \sum_{m=1}^{M} x_{i^*(z_{S-1},m)}, \tag{36}$$

where

$$i^*(z_{S-1}, m) = \underset{i}{\operatorname{argmax}} - \left(\|z_{S-1} - x_i\|^2 + \sigma u_{i,m}\right)$$
$$= \underset{i}{\operatorname{argmax}} - \left(\frac{1}{\sigma}\|z_{S-1} - x_i\|^2 + u_{i,m}\right)$$

If $u_{i,m} \sim$ Gumbel$(0,1)$, then by applying the Gumbel max-trick, we conclude that $i^*(z_{S-1}, m) \sim$ Categorical$(\pi_\sigma^i)$. This is a distribution over the indices $i = 1, ..., N$ of training samples, with event probabilities given by

$$\pi_\sigma^i = \operatorname{softmax}\left(-\frac{1}{\sigma}\|z_{S-1} - x_i\|^2\right)_i \tag{37}$$

If we represent $x_{i^*}$ as $X e_{i^*}$, where $X \in \mathbb{R}^{D \times N}$ is the matrix whose $i$-th column is training sample $x_i$ and $e_{i^*} \in \mathbb{R}^N$ is the $i^*$-th standard basis vector, then

$$z_S = \frac{1}{M} \sum_{m=1}^{M} x_{i^*(z_{S-1},m)}$$

$$= \frac{1}{M} \sum_{m=1}^{M} (X e_{i^*})$$

$$= \frac{1}{M} X \sum_{m=1}^{M} e_{i^*}$$

$$= \frac{1}{M} X I_\sigma$$

But $I_\sigma := \sum_{m=1}^{M} e_{i^*}$ is a realization of a Multinomial$(\pi_\sigma, M)$ random variable; this fact completes the proof of Proposition B.1.

This result further highlights the role of the smoothing parameter $\sigma$ in determining the distribution of a smoothed CFDM's samples: It is the *temperature* of the softmax determining $\pi_\sigma^i = $ softmax $\left(-\frac{1}{\sigma}\|z_{S-1} - x_i\|^2\right)$. When $\sigma = 0$, the softmax simply picks out the training sample $x_i$ that is closest to $z_{S-1}$. Conversely, as $\sigma \to \infty$, the event probabilities $\pi_\sigma^i$ become uniform and $z_S$ becomes the barycenter of $M$ uniformly-sampled training points.

## C  DETAILS ON NEAREST-NEIGHBOR ESTIMATOR OF CLOSED-FORM SCORE

Karppa et al. (2022) propose an unbiased estimator of a kernel density estimate $KDE(z)$. Given a kernel function $K_h(z)$ with bandwidth $h > 0$ and a dataset $\{x_i\}_{i=1}^N$, their estimator first searches for the $K$-nearest neighbors $\{x_k\}_{k=1}^K$ of $z$ in the dataset, then draws $L$ random samples $\{x_\ell\}_{\ell=1}^L$ from the remainder of the dataset, and approximates $KDE(z)$ as follows:

$$\widehat{KDE}(z) = \frac{1}{N} \sum_{k=1}^{K} K_h(x_k, z) + \frac{N-K}{LN} \sum_{\ell=1}^{L} K_h(x_\ell, z) \tag{38}$$

This estimator is unbiased for *any* subset of points $x_k \in \{x_i\}_{i=1}^N$ drawn in the first stage. In particular, using approximate nearest-neighbors (ANNs) rather than exact nearest neighbors of $z$ increases the variance of (38) but does not introduce bias.

As the closed-form score $\nabla \rho_t^*$ is the score of a Gaussian KDE $\rho_t^*$ with bandwidth $h = 2(1-t)^2$, we approximate the closed-form score using the following ratio estimator:

$$\nabla \widehat{\log \rho_t^*}(z) = \left( \frac{\widehat{\nabla \rho_t^*(z)}}{\rho_t^*(z)} \right) = \frac{\nabla \widehat{\rho_t^*}(z)}{\widehat{\rho_t^*}(z)}, \tag{39}$$

where $\widehat{\rho_t^*}(z)$ is Karppa et al. (2022)'s estimator (38). Since the gradient operator is linear, both the numerator and denominator in (39) are unbiased estimates of their respective terms in the closed-form score.

## D  ADDITIONAL EXPERIMENTAL DETAILS AND RESULTS

### D.1  PIXEL SPACE DDPM TRAINING DETAILS

We use $80\%$ of images for training and $20\%$ for testing. The backbone architecture for DDPM is a symmetric 2D UNet, a standard architecture utilized in leading models such as Stable Diffusion and based on the `diffusers` library. This UNet configuration incorporates both downsampling and upsampling 2D blocks, alongside self-attention blocks that make use of two ResNets per block. For the encoder, we implement four downsampling 2D blocks, followed by a self-attention block,

and conclude with a final downsampling 2D block. The decoder structure is symmetrical to that of the encoder. The input channel dimensions consist of three channels for all datasets, and the latent space dimensions increase from 128 to 256 to 512 every two blocks. In total, the model encompasses 113,673,219 model parameters. For training we use a standard DDPM scheduler with a time discretization of 1,000 for training and 100 for sampling. We train for 200 epochs, with a learning rate of $10^{-4}$ and a batch size of 16, using an accelerator with FP16 mixed precision. We normalize our images to lie in $[-1, 1]$ prior to training.

### D.2 CIFAR-10 LATENT SPACE GENERATION DETAILS

We perform diffusion in the 384-dimensional latent space of a pretrained autoencoder. The autoencoder consists of 2D convolutional layers and two linear layers with Gaussian Error Linear Unit (GELU) activations. The model has a total of 992,771 trainable parameters. We observe that, for meaningful generalization to occur when using $\sigma$-CFDMs, the autoencoder must sufficiently compress the data to prevent it from merely learning to linearly downsample the training images. Otherwise, sampling using a $\sigma$-CFDM in latent space would suffer from similar handicaps as when sampling directly in pixel space.

### D.3 MNIST

For the sake of completeness, we include some MNIST samples from our $\sigma$-CFDM in Figure 9. We sample from our model both directly in pixel space and within the 2-dimensional latent space of a MNIST autoencoder (with no KL regularization) based on 2D convolutions and transposed 2D convolutions. From the generated samples, it is evident that drawing $\sigma$-CFDM samples in a compressed latent space yields samples that are less noisy and neater. This can be attributed to the fact that it is easier to find sensible combinations of the observed samples in a lower-dimensional representation. Additionally, the decoder network aids in merging representations and correcting high-frequency details in the generated images. In Figure 10 we show samples generated by fitting a Gaussian distribution to the unregularized latent space. We can observe that many of the generations are noisy and the Gaussian distribution does not appropriately capture the learned latent structure.

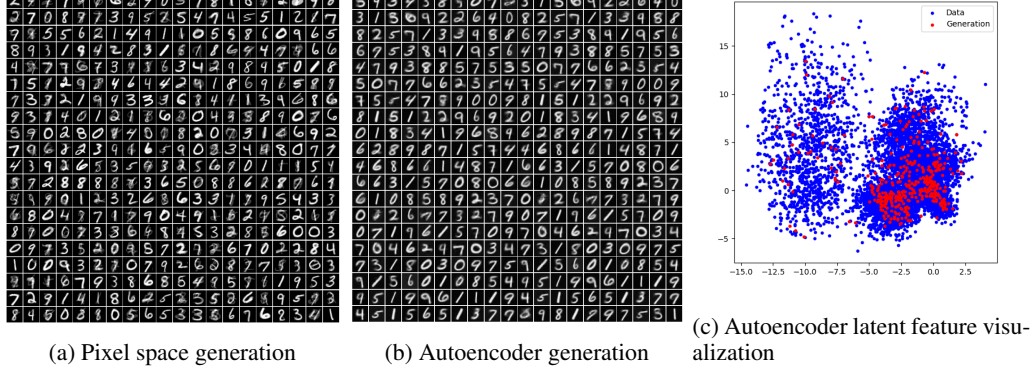

(a) Pixel space generation     (b) Autoencoder generation     (c) Autoencoder latent feature visualization

Figure 9: $\sigma$-CFDM results for MNIST, $T = 0.95$, $M = 4$, $h = 10^{-2}$, and $\sigma = 0.2$. The samples in (a) are generated directly in pixel space, whereas in (b) we sample in an autoencoder's 2-dimensional latent space. We plot the latent features in (c).

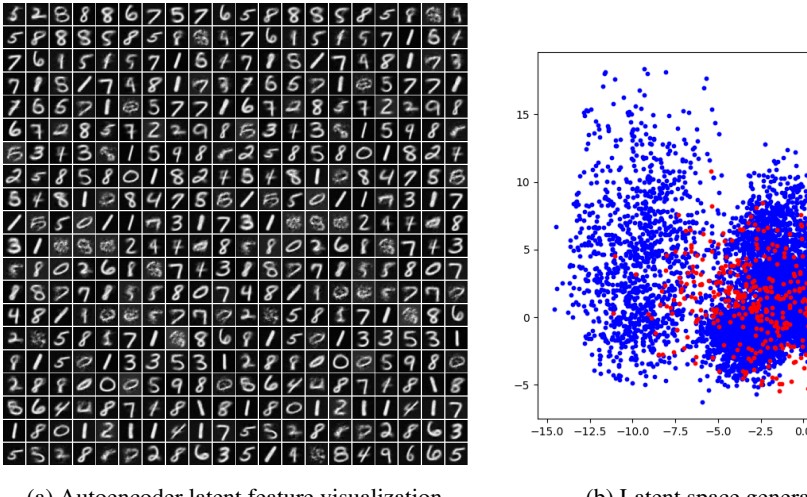

(a) Autoencoder latent feature visualization          (b) Latent space generation

Figure 10: Fitting a multivariate Gaussian to the MNIST autoencoder latent space. The multivariate Gaussian clearly fails to capture regions of the original distribution. Many of the generated samples are noisy.

### D.4 ONE-STEP GUMBEL SAMPLING

In this appendix, we implement and experiment with an alternative *one-step sampler* based on Proposition B.1, which characterizes a $\sigma$-CFDM's model distribution when starting at the final Euler iteration in Algorithm 1 and perturbing the distance weights in (4) with iid Gumbel noise.

Following the notation of Proposition B.1, we initialize this sampler with samples of the form $z_{S-1} = x_i + 0.2\epsilon$, where $\epsilon \sim \mathcal{N}(0, I)$; we found that perturbing the training points $x_i$ with higher-variance Gaussian noise than called for by the unsmoothed CFDM distribution $\rho^*_{t_{S-1}}$ improves the diversity of this method's samples. In each case, we directly perturb the distance weights in (4) with $u_{i,m} \sim \text{Gumbel}(0,1)$, and we report the values of $\sigma$ used to generate each set of samples in Figure 11. We accelerate our sampler and mitigate memory issues using the nearest-neighbor-based approximations described in Section 5.3.

As described in Proposition B.1, in the limit of small step sizes, this method yields samples of the form $z_S = \frac{1}{M} X I_\sigma$, where $X \in \mathbb{R}^{D \times N}$ is the matrix whose $i$-th column is training sample $x_i$ and $I_\sigma \sim \text{Multinomial}(\pi_\sigma, M)$. The probability $\pi^i_\sigma$ of training point $x_i$ is given by $\pi^i_\sigma = \text{softmax}\left(-\frac{1}{\sigma}\|z_{S-1} - x_i\|^2\right)$. Intuitively: This method is equivalent to drawing query points $z_{S-1}$ that are near the support of the data distribution $\rho_1$, drawing $M$ training points $x_i$ with probabilities $\pi^i_\sigma = \text{softmax}\left(-\frac{1}{\sigma}\|z_{S-1} - x_i\|^2\right)$, and then returning their barycenters.

This sampler achieves significantly higher sample throughput ($\sim 4$ times faster for CIFAR generation in latent space) than the $\sigma$-CFDM that we experiment with in Section 6.4. While the decoded samples are reasonable and perceptual image quality metrics are comparable to Section 6.4, the CIFAR-10 model samples in Figure 11 retain the appearance of superpositions of training samples to a significantly greater degree than in Section 6.4.

| Method | Metric | Butterflies (128 x 128) | CIFAR-10 (latents) |
|:---:|:---:|:---:|:---:|
| | LPIPS ↓ | $0.38 \pm 0.00$ | $0.42 \pm 0.00$ |
| | KID ↓ | $0.02 \pm 0.00$ | $0.02 \pm 0.00$ |
| One-step Gumbel | Training Time | $0 \pm 0$ h | $0 \pm 0$ h |
| | CPU (x86-64) Generation | $2.39 \pm 0.71$ s | $0.04 \pm 0.03$ s |

Table 2: Metrics for sample quality and generation time. CPU generation times are based on batches of 25 images.

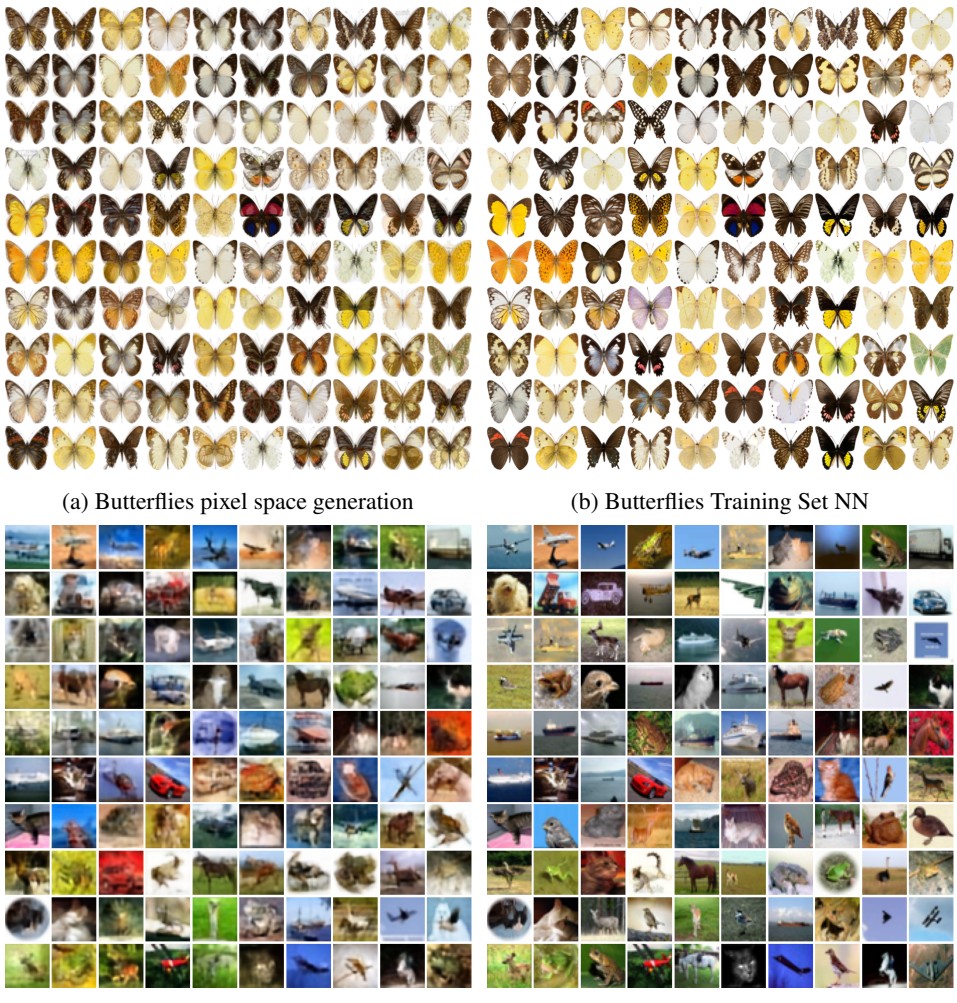

(a) Butterflies pixel space generation

(b) Butterflies Training Set NN

(c) CIFAR-10 latent space generation

(d) CIFAR-10 training set NN

Figure 11: One-step Gumbel model samples with $T = 0.98$, $M = 4$, and $\sigma = 0.3$ for Butterflies and $T = 0.98$, $M = 2$, and $\sigma = 0.2$ for CIFAR-10.

# E IMPACT OF $M$ ON MODEL SAMPLES

In this appendix, we demonstrate the impact of $M$ – the number of noise samples used to computed the smoothed score (3) – on a $\sigma$-CFDM's model samples. In Figure 12, we use a simple training set of 2 points (in blue), fix $\sigma = 1$, generate 100 $\sigma$-CFDM samples (in red) for different values of $M$. Note in particular that for large values of $M$, the model samples cluster around the centroid of the two training points. We conjecture that this phenomenon may be explained by the law of large numbers: As $M \to \infty$, $\frac{1}{M} \sum_{m=1}^{M} k_t(x + \sigma\epsilon_m) \to \mathbb{E}_\epsilon k_t(x + \sigma\epsilon)$, which is a deterministic quantity lying on the line segment connecting the two training points. In this regime, the reasoning used in the proof of Theorem 5.1 suggests that conditional on the second-to-last sampling iterate $z_{S-1}$, the output of a $\sigma$-CFDM becomes deterministic and all randomness in the model samples originates from $z_{S-1}$.

In Figure 13, we carry out a similar experiment with a training set consisting of 500 samples from the checkerboard distribution and $\sigma = 0.3$. Note that for large values of $M$, the model samples recede from boundary of the convex hull of the training data; we conjecture that this is an instance of the same phenomenon as in Figure 12.

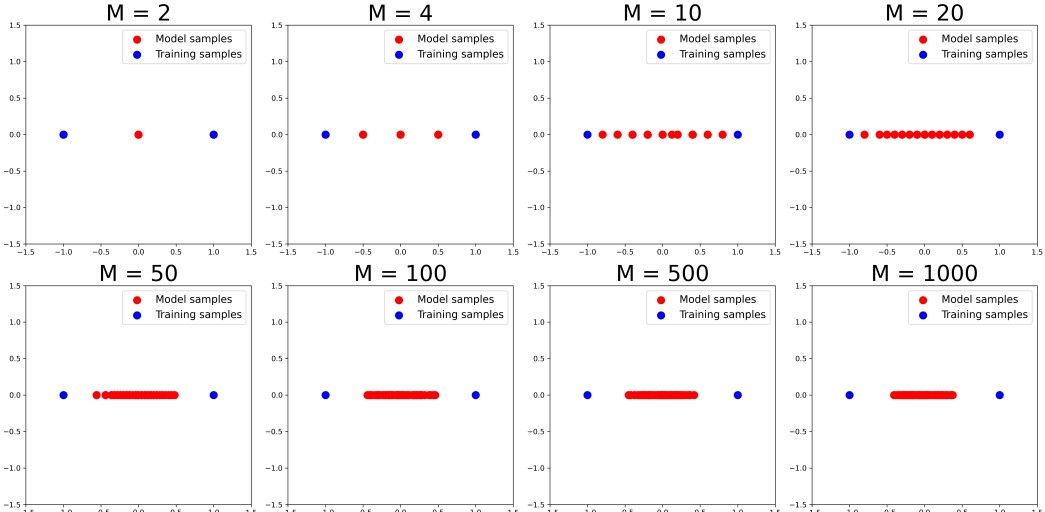

Figure 12: $\sigma$-CFDM samples (in red) generated given two training points (in blue) for various $M$.

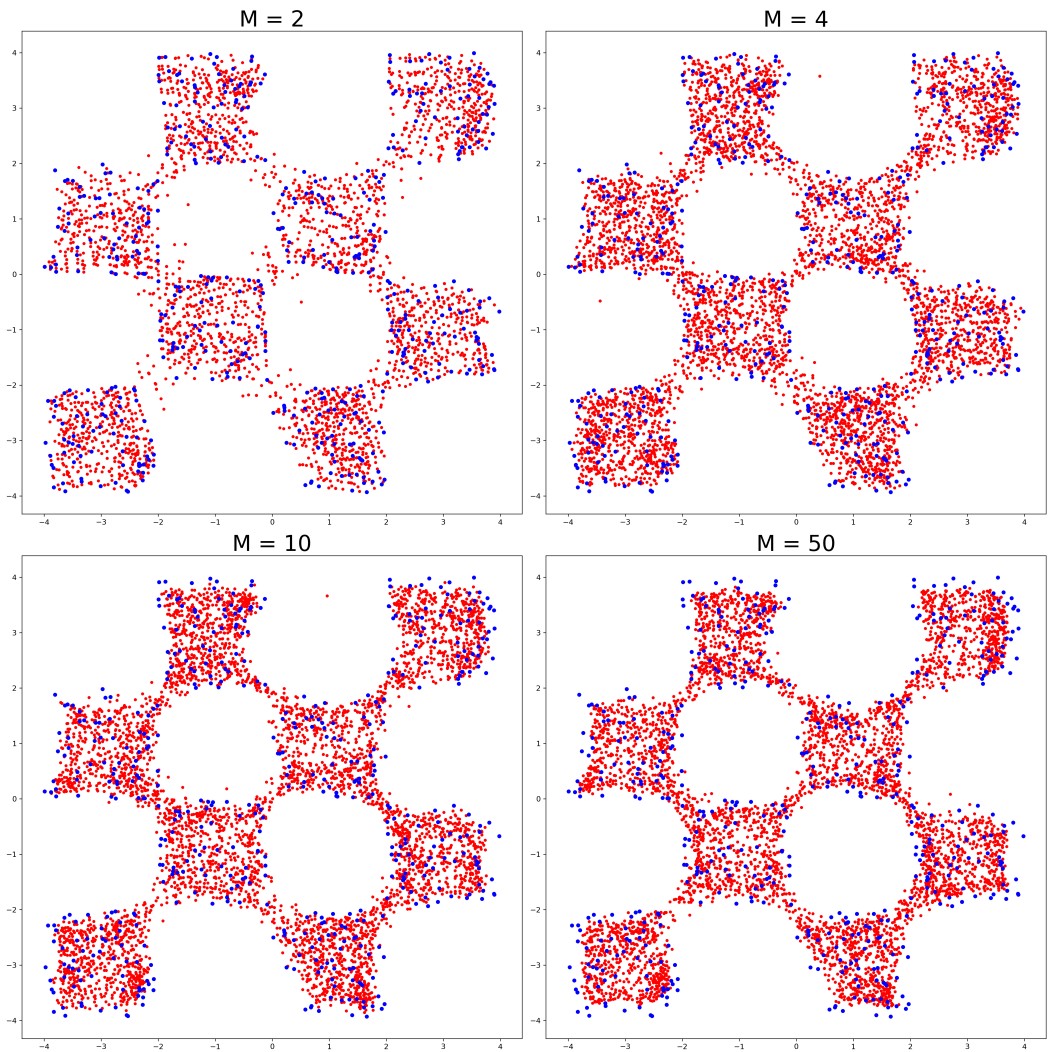

Figure 13: $\sigma$-CFDM samples (in red) generated given 500 training samples from the checkerboard distribution (in blue) for various $M$.

