# OpenReview forum: "Closed-Form Diffusion Models"
_ICLR.cc/2024/Conference — Submitted to ICLR 2024_

### Official Review · Reviewer_pdyz · 2023-10-30

**Soundness:** 3 good
**Presentation:** 4 excellent
**Contribution:** 3 good
**Rating:** 6
**Confidence:** 4

**Summary:**

This paper introduces closed-form diffusion models by smoothing the closed-form score function with finite training data. By explicitly introducing error through this smoothing process, the resulting diffusion models, referred to as $\\sigma$-CFDM, exhibit generalization capabilities. The paper also provides proof that $\\sigma$-CFDM's support contains the exact barycenters of $M$-tuples from the training points. To expedite the sampling process, the authors propose techniques such as initializing with unsmoothed CFDM samples at $T>0$ and utilizing an approximate nearest neighbor search method. Notably, experiments illustrate that $\\sigma$-CFDM can generate novel samples without the need for a training stage.

**Strengths:**

Originality: Although Equation (1,2) has been referenced in several existing works, the extension of this concept to closed-form diffusion models with generalization capabilities is an intriguing innovation. The deliberate introduction of error through a smoothing process not only distinguishes this work but also explicitly defines the inductive bias, a valuable contribution in the current deep learning-dominated era.

Quality: The work is well-motivated and logically compact. The presented propositions and the theorem are solid.

Clarity: The paper is generally well-written and easy to follow.

Significance: The paper offers a fresh perspective on the study of diffusion models. However, some concerns about its overall significance are detailed in the following section.

**Weaknesses:**

My primary concern about this paper revolves around the apparent simplicity of the generalization achieved by $\sigma$-CFDM. Indeed, the generalization abilities of generative models trained on finite datasets inherently arise from their deviation from a precise fit to the empirical distribution. Introducing explicit error to the empirical distribution, as presented in this paper, seem more elegant and interpretable compared to conventional deep learning methods. However, it's important to note that the proposed smoothing method essentially makes an assumption on the underlying data distribution (or the inductive biases). It posits that the barycenters of empirical data points also reside within the supports of the true data distribution. This assumption shares similarities with the mixing of training data points. An alternative approach would be to directly sample the mixup, rendering the diffusion process unnecessary. For instance, a generative model could be defined by initially sampling $M$ training data points and then returning their barycenter (with probabilities related to the variance) as the generated sample. Alternatively, one could employ a weighted average by further sampling weighting parameters from the $(M-1)$-simplex. It would be valuable to provide a comparative analysis with these alternative methods to offer a more comprehensive evaluation (e.g. FID score and visualization). As the authors mention in Section 6.4, for image data, the barycenters may not be well-registered, and an auto-encoder is adopted, such a comparison should also be conducted in the latent space for real image generation.

**Questions:**

- How does $M$ affect the behavior of $\\sigma$-CFDM?
- Can this method be extended to conditional generation?

Please also see my questions in the Weaknesses section.

---

> ### Author Response · Authors · 2023-11-15
> **Response to Official Review of Submission5670 by Reviewer pdyz**
>
> We thank the reviewer for their insightful comments, which will improve the quality of our manuscript, and are glad they share our interest in closed-form diffusion models that generalize. We have uploaded a revision to OpenReview, where the reviewer will find that we have incorporated many of their suggestions. In this revision, we have typeset our relevant changes in teal. We respond to the reviewer's comments below.
>
> **Inductive bias of our method.**
>
> As the reviewer points out, our method assumes that the barycenters of nearby training points lie within the support of the data distribution. This is a reasonable prior if the data distribution is supported on a manifold whose curvature isn’t too large relative to the number of training samples. As a classical example, Roweis and Saul’s “locally linear embedding” imposes a very similar prior on its inputs.
>
> While this assumption often isn’t true of the image manifold in pixel space (barycenters of images typically don’t look like natural images), by mapping our training data to a latent space with an appropriately-regularized autoencoder, we obtain a latent data manifold whose structure is more favorable to our method. This explains why our method is able to generate plausible CIFAR samples when operating in the latent space of an autoencoder with a tight bottleneck. We have updated our discussion in Section 6.4 to clarify this point.
>
> For future work, we will explore training an autoencoder that enforces local linearity in the latent space; see e.g. Section 4.2 in “Learning Signal-Agnostic Manifolds of Neural Fields” by Du et al. (2021). We hope this strategy will yield a latent data manifold with a locally-linear structure that is highly favorable to our method and produce natural-looking decoded samples. We have updated our conclusion to include this future direction.
>
> **Alternative sampling processes.**
>
> Naively-implemented “generative mixup” would draw pairs of training points uniformly from the training set and then output some data-independent convex combination of these points. This amounts to sampling from the convex hull of the training set, and is not a suitable inductive bias for most target distributions. One exception is when the target distribution is log-concave; in this case, convex combinations of pairs of target samples have log-likelihood at least as large as one of the endpoint samples. For example, all Gaussian distributions are log-concave. Our fitted Gaussian baseline in Section 6.4 (which performs very poorly) demonstrates that this log-concavity prior is unsuitable for image generation.
>
> On the other hand, directly sampling tuples of $M$ training points with probability depending on their variance and returning their barycenters is challenging, because there are $N^M$ such tuples for training sets of size $N$.
>
> Instead, we have implemented and experimented with a new sampler in Appendix D.4 of the revision, which implements the one-step sampling procedure described in Proposition B.1. This is an extreme case of our existing strategy for reducing the number of sampling steps using a warm start from an unsmoothed CFDM, and uses Gumbel perturbations $u_{i,m}$ to the distance weights in Equation (4) for analytical convenience. Proposition B.1 shows that this method is equivalent to drawing a batch of query points $z_{S-1}$ lying near the support of the data distribution $\rho_1$, drawing $M$ training points $x_i$ with probabilities  $\pi^i_\sigma = \textrm{softmax}\left(-\frac{1}{\sigma} \|z_{S-1} - x_i \|^2 \right)$, and then returning their barycenters.
>
> As expected, this method achieves significantly higher sample throughput than our existing approach ($\sim 4$ times faster for CIFAR generation in latent space). While the decoded samples are reasonable, the speedup comes at a cost to sample quality: Many of our decoded samples retain the appearance of superpositions of training samples.
>
> **Effect of $M$ on CFDM samples.**
>
> We have included an experiment in Appendix E of the revised manuscript demonstrating the impact of $M$ on our method’s samples.
>
> **Extension to conditional generation.**
>
> In future work, we intend to explore using our method for conditional generation by using Dhariwal and Nichol (2021)'s classifier guidance, which would amount to augmenting our velocity field (7) with the gradient of a pretrained classifier. We have updated our conclusion to include this future direction.
>
> **Conclusion.**
>
> We thank the reviewer for their comments and hope that we have adequately addressed their concerns. If they are satisfied with our answers and our revision, we respectfully request that they raise their score for this paper. Otherwise, we would be pleased to continue this discussion during the reviewer-author discussion period.

---

> > ### Comment · Reviewer_pdyz · 2023-11-21
> >
> > Thank you for your response. I appreciate the additional insights into the inductive bias, although I still hold the view that the introduced inductive bias might be somewhat trivial. Regarding the convex hull, I believe $\sigma$-CFDM also samples from the convex hull, so I don't see a clear drawback of "naive mixup generation" compared to $\sigma$-CFDM. However, I appreciate the idea of building generative models in a nonparametric sense, where introducing an effective and nontrivial inductive bias is both challenging and intriguing. Therefore I would like to maintain my original rating.

---

> > > ### Author Response · Authors · 2023-11-22
> > > **Final remarks for Reviewer pdyz**
> > >
> > > Thank you for your response. Theorem 5.1 in our paper indeed shows that our method samples from the convex hull of the training data. The difference between our method’s samples and the expected output of “naive mixup generation” is illustrated in Figure 2 of our paper. Naive mixup would output samples like those on the right side of Figure 2, where the sample distribution is effectively uniform over the convex hull of the training points.
> > >
> > > On the other hand, by varying $\sigma$, our model distribution can smoothly interpolate between the empirical distribution over training data (left side of Figure 2), a distribution with support throughout the data manifold (middle of Figure 2), and in the large $\sigma$ limit, a distribution that is nearly uniform over the convex hull of the training data. In this way, one can view naive mixup as a special case of our model, where $\sigma$ is set too large to be practically useful.
> > >
> > > We are pleased that you appreciate the innovativeness of our closed-form approach to generative modeling and share our excitement for generative modeling with minimal neural computation. We view your interest in our approach as an indication that our work will be a thought-provoking part of this year’s ICLR program, where we hope that it will inspire followup work exploring alternative inductive biases.

---

> ### Author Response · Authors · 2023-11-20
> **Reminder for Reviewer pdyz**
>
> Dear Reviewer pdyz -- as the discussion period will close in a few days, we would greatly appreciate if you would take a look at our response to your review soon and let us know if you would like to see further changes to our manuscript. We look forward to addressing your remaining concerns before the end of discussion period. If our response was satisfactory, we ask that you consider raising your score for our submission. Thank you for your time.

---

### Official Review · Reviewer_P7kJ · 2023-10-31

**Soundness:** 3 good
**Presentation:** 2 fair
**Contribution:** 3 good
**Rating:** 5
**Confidence:** 3

**Summary:**

The paper considers smoothed closed-form diffusion models. In particular, the authors study the properties of smoothed score function and propose a new sampling algorithm. Numerical experiments are organized to evaluate the performance of this new model.

**Strengths:**

1. The authors propose a new diffusion model with closed-form score function.
2. The paper studies analytic properties of smoothed diffusion models, including support of output distribution and the approximation error.
3. Comprehensive numerical experiments are organized. The smoothed diffusion model is compatible with neural network-based diffusion models such as DDPM.
4. The model is training-free. Sampling can be implemented even in CPUs.

**Weaknesses:**

1. If my understanding is correct, the motivation is to have a diffusion model with good generalization capacity. Although the smooth diffusion model performs as well as DDPM, it is still unclear how it is connected to the generalization of diffusion models.
2. In terms of modeling, the only novelty seems to be an additional smooth term added to $k$. Could you point out your contribution more clearly?
3. The writing looks good but can still be improved.

**Questions:**

I have some minor questions:

1. Why do we have a closed-form score function as in Section 3? Could you give some references or provide derivations in the appendix?
2. Seemingly, you only provide a comparison to DDPM in terms of training time and LPIPS. Could you provide more comprehensive experiments compared to other benchmarks and additional metrics?

---

> ### Author Response · Authors · 2023-11-15
> **Response to Official Review of Submission5670 by Reviewer P7kJ**
>
> We thank the reviewer for their comments, which we believe will improve the quality of our manuscript. We have uploaded a revision to OpenReview, where the reviewer will find that we have incorporated many of their suggestions. In this revision, we have typeset our relevant changes in teal. We respond to the reviewer's comments below.
>
> **Relationship to the generalization of diffusion models.**
>
> Our goal is to design a diffusion model that generalizes controllably while requiring no training and being efficient to sample. While our method is indeed inspired by recent work studying the generalization of diffusion models, our contribution is not a rigorous study of the generalization of existing (neural) diffusion models, but rather introducing a new class of diffusion models that do not require trained approximations to the score function.
>
> In particular: Pidstrigach (2022) shows that a score-based generative model (SGM) can generalize only if the model score incurs unbounded approximation error relative to the true score; otherwise, the model memorizes its training data. We combine this strong result with the observation that neural networks tend to learn overly smooth approximations to their target function and propose a simple, training-free method to induce the error that Pidstrigach (2022) shows to be necessary, in a form similar to that of neural SGMs.
>
> We have updated our introduction and our lead-in to Section 4 to clarify these points.
>
> **Clarifying our contribution.**
>
> In addition to smoothing the closed-form score and showing that this promotes generalization, we introduce two approximation techniques to render this method tractable: Initializing our sampler with unsmoothed CFDM samples at $T>0$ and a nearest-neighbor-based score approximation. These approximations enable our method to achieve very fast inference speeds (for example, 138 latents/sec on our CIFAR experiments) while running on a consumer-grade laptop without a dedicated GPU.
>
> We have updated our introduction to clarify this point.
>
> **References for closed-form score formula in Section 3.**
>
> We have updated our discussion in Section 3 to include several references for this formula. In particular, Appendix B.3 of “Elucidating the Design Space of Diffusion-Based Generative Models” by Karras et al. (2022) includes a full derivation of the closed-form score formula; their “ideal denoiser” corresponds to our function $k_t$.
>
> **Additional benchmarks and metrics.**
>
> In our revised manuscript (Table 1), we have also reported our image generation results in terms of the kernel inception distance (KID). Furthermore, in Appendix D.4 of the revision, we implement and experiment with an alternative sampler. This sampler implements the one-step procedure described in Proposition B.1 and is an extreme case of our existing strategy for reducing the number of sampling steps, using a warm start from an unsmoothed CFDM and Gumbel perturbations to the distance weights in Equation (4) for analytical convenience. Proposition B.1 shows that this equivalent to drawing a batch of query points $z_{S-1}$ lying near the support of the data distribution $\rho_1$, drawing $M$ training points $x_i$ with probabilities  $\pi^i_\sigma = \textrm{softmax}\left(-\frac{1}{\sigma} \|z_{S-1} - x_i \|^2 \right)$, and then returning their barycenters.
>
> As expected, this method achieves significantly higher sample throughput than our existing approach ($\sim 4$ times faster for CIFAR generation in latent space). While the resulting outputs are reasonable, the speedup comes at a cost to sample quality: Many of our decoded samples retain the appearance of superpositions of training samples.
>
> **Conclusion.**
>
> We thank the reviewer for their comments and hope that we have adequately addressed their concerns. If they are satisfied with our answers and our revision, we respectfully request that they raise their score for this paper. Otherwise, we would be pleased to continue this discussion during the reviewer-author discussion period.

---

> > ### Comment · Reviewer_P7kJ · 2023-11-21
> > **Response to authors**
> >
> > Thanks to the authors for the revision of this paper. Most of my concerns have been well-addressed. However, I also agree with Reviewer sZVo that additional baselines and metrics should be included. The experiments in the revision are much better compared to the original version but still not enough for acceptance. Also, the proposed model seems to be a little improvement to the existing work. Unfortunately, in this situation, I believe my evaluation is fair.

---

> > > ### Author Response · Authors · 2023-11-22
> > > **Final remarks for Reviewer P7kJ**
> > >
> > > Thank you for your response. We are pleased to have addressed most of your concerns in our revision. Including additional baselines and metrics would not substantively change the conclusions of our work. Our contributions are as follows:
> > >
> > >
> > > - We show that smoothing the exact solution to the score-matching problem promotes generalization.
> > > - Using our smoothed score, we construct a closed-form sampler that generates novel samples without requiring any training, and characterize the support of its samples.
> > > - We accelerate our sampler using a nearest-neighbor based estimator of our smoothed score and by taking fewer sampling steps. We show that in practice, one can aggressively approximate our smoothed score at little cost to sample accuracy.
> > > - We scale our method to high-dimensional tasks such as image generation. By operating in the latent space of a pretrained autoencoder, we generate good-quality, novel samples from CIFAR-10 at a rate of 138 latents per second on a consumer-grade laptop with no dedicated GPU.
> > >
> > >
> > > If any of these claims are poorly-supported, we would be happy to bolster them with further experiments. We welcome any specific hypotheses related to our contributions that the reviewer would like to see experimentally validated.

---

> ### Author Response · Authors · 2023-11-20
> **Reminder for Reviewer P7kJ**
>
> Dear Reviewer P7kJ -- as the discussion period will close in a few days, we would greatly appreciate if you would take a look at our response to your review soon and let us know if you would like to see further changes to our manuscript. We look forward to addressing your remaining concerns before the end of discussion period. If our response was satisfactory, we ask that you consider raising your score for our submission. Thank you for your time.

---

### Official Review · Reviewer_sZVo · 2023-10-31

**Soundness:** 2 fair
**Presentation:** 3 good
**Contribution:** 3 good
**Rating:** 5
**Confidence:** 5

**Summary:**

The paper considers the score-based generative models (SGMs) - precisely, their ability to generating novel data and prevent from the memorization the training data. The usual solution is approximating the score training a neural network via score-matching. Despite promoting the generalization, the neural SGMs are costly to train and sample. Instead, the authors propose to explicitly smooth the closed-form score to obtain an SGM that generates novel samples without training. In this work, they also formulate an efficient k-NN based estimator of their score function.

**Strengths:**

The paper has a few significant strengths overall, which I will outline below:
1. The proposed method is easy, but elegant.
2. The significant advantage of the proposed method is the training and sampling times and the possibility to use the standard CPU.
3. The authors tested the method in different settings.
4. Overall, the flow of the manuscript is well-organized.

**Weaknesses:**

However, despite the strengths, the paper has a few major and minor weaknesses:
1. The method is compared only on a small resolution datasets - the largest resolution have the Butterflies dataset, which still is only 128x128 (and has small number of examples). I’m not sure if this model will be working well on a larger resolutions, e.g. at least 256x256 ImageNet. Maybe the comparable results to the DDPM is only a matter of not so large resolutions?
2. The authors compared their method only against the DDPM. Could you compare with different diffusion models, which might generalize better?
3. The presented results are not sufficient and unclear. They are unclear, because even in the Table 1, the proposed method is better on one dataset, whereas being worse on another. It would be helpful to include also other metrics than LPIPS (e.g., FID, SSIM).
4.  In the whole paper, the Figures and Tables are too small. It is very hard to see what is in the Table and if the proposed method is better than DDPM. The presented samples are also way too small - based on this presentation I just cannot compare the proposed model against DDPM.

**Questions:**

I would like to see especially the following experiments and improvements regarding specifically to the Weaknesses section:
1. Please if you could include comparison on datasets having higher resolutions, like ImageNet.
2. The comparison against others diffusion models (e.g., DDIM) is needed.
3. I would like to see results in other metrics also (like FID or SSIM).
4. Please if you be able to enlarge all the Figures and all the Tables.

---

> ### Author Response · Authors · 2023-11-15
> **Response to Official Review of Submission5670 by Reviewer sZVo**
>
> We thank the reviewer for their comments, which we believe will improve the quality of our manuscript. We have uploaded a revision to OpenReview, where the reviewer will find that we have incorporated many of their suggestions. In this revision, we have typeset our relevant changes in teal. We respond to their comments below.
>
> **Additional baselines.**
>
> In Appendix D.4 of the revision, we implement and experiment with an alternative sampler which serves as an additional baseline for our model. This sampler implements the one-step procedure described in Proposition B.1 and is an extreme case of our existing strategy for reducing the number of sampling steps using a warm start from an unsmoothed CFDM and Gumbel perturbations to the distance weights in Equation (4) for analytical convenience. Proposition B.1 shows that this equivalent to drawing a batch of query points $z_{S-1}$ lying near the support of the data distribution $\rho_1$, drawing $M$ training points $x_i$ with probabilities  $\pi^i_\sigma = \textrm{softmax}\left(-\frac{1}{\sigma} \|z_{S-1} - x_i \|^2 \right)$, and then returning their barycenters.
>
> As expected, this method achieves significantly higher sample throughput than our existing approach ($\sim 4$ times faster for CIFAR generation in latent space). While the resulting outputs are reasonable, the speedup comes at a cost to sample quality: Many of our decoded samples retain the appearance of superpositions of training samples.
>
> **Additional metrics.**
>
> In our revised manuscript, we have also reported our image generation results in terms of the kernel inception distance (KID).
>
> **Conclusion.**
>
> We thank the reviewer for their comments and hope that we have adequately addressed their concerns. If they are satisfied with our answers and our revision, we respectfully request that they raise their score for this paper. Otherwise, we would be pleased to continue this discussion during the reviewer-author discussion period.

---

> > ### Comment · Reviewer_sZVo · 2023-11-20
> >
> > I want to thank the Authors for their response and revision of the paper. I went through all the added changes in the manuscript, the response and additional experiments. Unfortunately, my concerns and indicated weaknesses weren’t properly addressed in the revised version of the paper.
> > In the following paragraphs, I will go through the specific points bolded in the authors response to my review.
> >
> >
> > **1. Additional baselines.**
> >
> >
> > The Authors propose to consider the additional sampler (being just an extreme case of their method) as another baseline. I agree that it is a good idea to include such a scenario in the paper - to see the limits of the proposed method. However, I asked exactly for comparison with another diffusion models frameworks (like DDIM) and to compare on the higher-dimensional data (e.g., ImageNet) being much closer to the current real-world problems in diffusion models. Unfortunately, the presented experiments and comparison with such baselines don’t answer the most important question to me, i.e., if the proposed method are able to scale to the higher-dimensional (more real) problems.
> >
> >
> > **2. Additional metrics.**
> >
> >
> > I appreciate and thank for adding the KID metric. However, it will be good to see also other metrics (like proposed FID and SSIM) to better comparison with the baselines.
> >
> >
> >
> > Overall, I thank the Authors for the response and including the revised version of the manuscript. I agree that the proposed method is interesting. However, I think that presented experiments do not show that the method is able to scale to higher-dimensional problems, at least 256x256 images (I asked for). The last I recognize as crucial for this paper. Otherwise, we don’t know the practical and theoretical limits of the proposed method. Unfortunately, in this situation, I cannot raise my score.

---

### Official Review · Reviewer_t4Fr · 2023-11-01

**Soundness:** 3 good
**Presentation:** 2 fair
**Contribution:** 1 poor
**Rating:** 5
**Confidence:** 4

**Summary:**

This paper proposes a new variant of diffusion models based on the fact that the score function can be equivalently written as the expectation over score functions of conditional distributions. This expectation can be written in closed-form as a weighted sum over all training points, thus the "closed-form" diffusion models. The paper then proposes to smooth the closed-form scores by integrating it over small noise perturbation of the inputs, akin to the denoising score matching approach. Due to the closed-form expression, sampling can be implemented without a parametric approximation to the score function. Because each evaluation of the score requires going through all training examples, a nearest-neighbor estimator is used to reduce the computation cost.

**Strengths:**

* The paper is clear and pleasant to read.
* The experimental study is carefully designed and investigated most questions that I could think of about the closed-form model.
* It's interesting that the proposed method can fill in the gaps between the sparse training samples in the 3D point cloud experiment (although it's unclear to me why the method was able to do so--see below).

**Weaknesses:**

* The proposed method is not well-positioned in literature. It's worth pointing out that the key idea of representing the marginal score as the expectation of scores of distributions conditioned on inputs is actually quite well-known. It has been used, for example, to develop the original denoising score matching objective [1]. It is also used in the literature as "score-interpolation" [2]. I just named a few but I would recommend the authors to do a thorough literature review as I believe this property is used in many more works.

* The definition of the notation \hat{c}_k (baycenters) is missing.

* The exponential dependence of the sampling error on T is concerning. Although empirical evidence is provided to justify that this error bound is pessimistic, it also renders the bound unnecessary. Meanwhile, it's unclear if the conclusion that under sigma < 0.4, a large starting T is harmless will generalize to other datasets.

* The 3D point cloud experiment is interesting but I don't understand why the proposed method fills the gap there. Could the authors elaborate on this?

* The practical utility of the proposed closed-form models is also unclear. Given that the model can only sample from baycenters of data point tuples, is there a clear case where we would prefer such a model over a trained score model?

* This is minor, but the readability of section 3 can be greatly improved if not going to the notation convention used by rectified flow, as the proposed method can be described using the standard diffusion model formulation (where the time is reversed and stochastic transition is used).

[1] Vincent, Pascal. "A connection between score matching and denoising autoencoders." Neural computation 23.7 (2011): 1661-1674.

[2] Dieleman, Sander, et al. "Continuous diffusion for categorical data." arXiv preprint arXiv:2211.15089 (2022).

**Questions:**

Please see questions above.

---

> ### Author Response · Authors · 2023-11-15
> **Response to Official Review of Submission5670 by Reviewer t4Fr**
>
> We thank the reviewer for their thoughtful comments, which will improve the quality of our manuscript. We have uploaded a revision to OpenReview, where the reviewer will find that we have incorporated many of their suggestions. In this revision, we have typeset our relevant changes in teal. We respond to the reviewer's comments below.
>
> **Positioning in the literature.**
>
> The closed-form expression for the score of $p_t$ (Equations (1) and (2) in our manuscript) is indeed well-known, with similar formulas having appeared in the empirical Bayes literature as early as 1961 in Miyasawa’s “An empirical Bayes estimator of the mean of a normal population.”
>
> We have updated Sections 1 and 3 to highlight this point and include several references to appearances of this formula in the statistics and ML literature.
>
> However, we emphasize that this formula is not one of our contributions. Rather, we use this formula as a starting point to develop a score-based generative model (SGM) that generalizes without requiring trainable neural approximations, and we introduce two strategies in Sections 5.2 and 5.3 (starting sampling at $T>0$ and the NN-based score estimator) that greatly accelerate our sampler.
>
> We have updated our statement of contributions in Section 1 to further emphasize this point.
>
> **Definition of notation $\bar{c}_k$ for barycenters.**
>
> We thank the reviewer for pointing out this oversight and have defined this notation in Section 4.2 of the revised manuscript.
>
> **Exponential dependence of the sampling error on $T$.**
>
> As the reviewer notes, the error bound that in Theorem 5.2 is pessimistic in its dependence on $T$. Given that our experiments in Section 6.2 show little accuracy loss for practical values of $\sigma$, we believe this bound to be loose as a consequence of the elementary techniques used to prove it. We would be interested in alternative methods that lead to an error bound which reflects the small error that we observe in practice.
>
> However, the bound in Theorem 5.2 also indicates that the error incurred by starting at $T>0$ scales linearly with $\sigma$. When $\sigma=0$, the smoothed and unsmoothed CFDMs are identical, and one can sample the unsmoothed CFDM (i.e. mixture of Gaussians) at any time without incurring error. For $\sigma>0$, this is no longer true – but it is not obvious how the error incurred by sampling the unsmoothed CFDM at $T>0$ depends on $\sigma$. Our result confirms that this dependence is linear, so small values of $\sigma$ lead to small errors.
>
> We have updated our discussion in Section 5.2 to clarify this point, namely that the interest of the theorem is the $\sigma$ dependence rather than the $T$ dependence.
>
> **Generality of conclusion that under $\sigma<0.4$, a large starting $T$ is harmless.**
>
> This was true across all datasets considered in this paper—not just the checkerboard dataset used in Section 6.2.
>
> For example, in Sections 6.3 and 6.4, we used a start time of $T=0.98$ for all image generation experiments with little degradation in sample quality. The fact that $[0,0.4]$ is the appropriate range for $\sigma$ is partially a consequence of having rescaled the training images/latents to lie within the unit ball before sampling and then reversed this rescaling on the generated samples. We have updated our discussion in Section 6.2 to clarify this point.
>
> Subject to time constraints within the discussion period, we would be happy to run any further experiments the reviewer has in mind to further confirm this observation.
>
> **Filling in gaps between sparse manifold samples in the 3D point cloud experiment.**
>
> We have updated our discussion of this experiment in Section 6.1 to clarify why our method fills in gaps between sparse samples from the surface.
>
> **Practical utility of CFDMs.**
>
> The benefits of our method over a neural SGM are twofold:
>
> 1. Computationally: Our method requires no training and can be sampled very quickly while running on consumer-grade CPUs.
> 2. Theoretically: Unlike with neural SGMs, we are able to characterize the support of our model’s samples (Theorem 5.1) and can read off the weights of the training points that contributed to each generated sample.
>
> As the reviewer highlights, our model samples barycenters of training points. In light of this fact, sampling in an appropriately-structured latent space (one where the data manifold is “locally linear”) is likely the most realistic path towards obtaining sample quality comparable to neural SGMs on high-resolution image generation tasks. Our CIFAR results in Section 6.4 indicate that this is a promising direction, and we continue to work toward scaling our method up to larger and higher-resolution datasets.
>
> **Conclusion.**
>
> We thank the reviewer for their comments and hope that we have addressed their concerns. If they are satisfied with our answers and our revision, we respectfully request that they raise their score. Otherwise, we would be pleased to continue this discussion.

---

> ### Author Response · Authors · 2023-11-20
> **Reminder for Reviewer t4Fr**
>
> Dear Reviewer t4Fr -- as the discussion period will close in a few days, we would greatly appreciate if you would take a look at our response to your review soon and let us know if you would like to see further changes to our manuscript. We look forward to addressing your remaining concerns before the end of discussion period. If our response was satisfactory, we ask that you consider raising your score for our submission. Thank you for your time.

---

### Author Response · Authors · 2023-11-16
**Changes to manuscript are typeset in teal**

Dear reviewers -- in our newly-uploaded revision, we have typeset the changes to the main body of our manuscript in **teal** for your convenience. We have also added two new appendices: Appendix D.4: "One-Step Gumbel Sampling" and Appendix E: "Impact of $M$ on Model Samples." Please don't hesitate to let us know if you have any further questions or concerns.

---

### Meta-Review · Area_Chair_EneZ · 2023-12-11

**Metareview:**

The paper proposes a closed form smoothed score function based on the training examples , that can be used in score based samplers as a generative model. Authors propose speedups for sampling via a nearest neighbor scheme and via working in a latent space of an auto-encoder and decoding back to the ambient space. The underlying assumption in this work is that the barycenters of training points (or their latent representation) remain on the data manifold. While this assumption may not hold on the data distribution it may hold in the latent domain.

Reviewers appreciated the simplicity and the elegance of this known training free score function and all the speedups the authors come up with to make sampling feasible. Two theoretical concerns were brought up by the reviewer:  on the exponential dependency of sampling error on $T$ and how to improve it to make sure that the sampling converges in finite time. The other one is on the assumption on the data distribution (in ambient or in latent space), how to ensure it allows sampling on data manifold using barycenters of points or their latent. Authors suggested a new work on autoencoders by Du et al that enforces local linearity. This can be a great fix for this issue.

There were also concerns regarding the evaluation metrics used, authors added in the rebuttal KID, reviewers suggested other metrics to be more comparable with the literature such as FID and SSIM. Some other suggestions by the reviewers were in including other diffusion baselines and on larger resolution datasets. Authors supplied a baseline that is an extreme case of their method and not from the literature.

The paper revision improved overall a lot the manuscript and the authors made an honest effort in responding to reviewer concerns.  Tightening the sampling error dependency on T, and making sure that the data distribution assumption is maintained in a latent space, will strengthen more the paper and make it ready for publication.

**Justification For Why Not Higher Score:**

The paper needs to address some concerns in the theory and in  experiments.

**Justification For Why Not Lower Score:**

N/A

---

### Decision · Program_Chairs · 2024-01-16

Reject